# Knowledge-Empowered Dynamic Graph Network for Irregularly Sampled Medical Time Series

**Yicheng Luo, Zhen Liu\*, Linghao Wang, Junhao Zheng, Binquan Wu, Qianli Ma**\*
School of Computer Science and Engineering,
South China University of Technology, Guangzhou, China
{csluoyicheng2001, cszhenliu, cskyun_ng}@mail.scut.edu.cn,
{linghaowang6, junhaozheng47}@outlook.com, qianlima@scut.edu.cn

## Abstract

Irregularly Sampled Medical Time Series (ISMTS) are commonly found in the healthcare domain, where different variables exhibit unique temporal patterns while interrelated. However, many existing methods fail to efficiently consider the differences and correlations among medical variables together, leading to inadequate capture of fine-grained features at the variable level in ISMTS. We propose Knowledge-Empowered Dynamic Graph Network (KEDGN), a graph neural network empowered by variables' textual medical knowledge, aiming to model variable-specific temporal dependencies and inter-variable dependencies in ISMTS. Specifically, we leverage a pre-trained language model to extract semantic representations for each variable from their textual descriptions of medical properties, forming an overall semantic view among variables from a medical perspective. Based on this, we allocate variable-specific parameter spaces to capture variable-specific temporal patterns and generate a complete variable graph to measure medical correlations among variables. Additionally, we employ a density-aware mechanism to dynamically adjust the variable graph at different timestamps, adapting to the time-varying correlations among variables in ISMTS. The variable-specific parameter spaces and dynamic graphs are injected into the graph convolutional recurrent network to capture intra-variable and inter-variable dependencies in ISMTS together. Experiment results on four healthcare datasets demonstrate that KEDGN significantly outperforms existing methods. Our code is available at `https://github.com/qianlima-lab/KEDGN`.

## 1 Introduction

In the medical environment, the widely used Electronic Health Records (EHRs) have abundant typical Irregularly Sampled Medical Time Series (ISMTS) data [1]. Each ISMTS typically comprises multiple medical variables for a patient, each with distinct medical properties, resulting in significant differences in the sampling patterns of each variable series. Additionally, due to the dynamic changes in a patient's condition, the sampling rate of variables varies over different periods, resulting in uneven sampling intervals [2].

Many existing methods for ISMTS primarily focus on addressing uneven sampling intervals and have proposed approaches such as Ordinary Differential Equations (ODEs) [3, 4] and continuous-time embeddings [5, 6], etc, which have already achieved significant success. Recent advancements in regularly sampled multivariate time series analysis [7, 8] underscore the importance of capturing variable-specific temporal patterns. However, many existing methods for ISMTS have not adequately considered this aspect, having limited ability to explicitly distinguish multiple variable series with

---

*Corresponding authors

38th Conference on Neural Information Processing Systems (NeurIPS 2024).

different time patterns and thus lacking finer-grained capturing of variable-level features. Particularly in ISMTS, different variables have distinct medical properties, further intensifying the degree of differences among variables. In such cases, capturing differentiated variable patterns requires a deeper exploration of inherent differences among variables.

Despite the differentiated temporal patterns occurring among variables within ISMTS, they are not entirely independent but exhibit medical correlations. Due to the dynamic changes in the patient's condition, this correlation varies along with the sampling density of variables at different periods, as illustrated in Figure 1. Some existing work has introduced graph neural networks [9, 10] to model the time-varying correlations among variables in ISMTS. However, due to the lack of prior medical knowledge, these methods learn variable correlation graphs from misaligned and imbalanced observations in variables of ISMTS and rely solely on downstream tasks for graph optimization. Consequently, the variable graphs learned by these methods may face challenges in accurately reflecting the general medical correlations among variables, resulting in suboptimal performance.

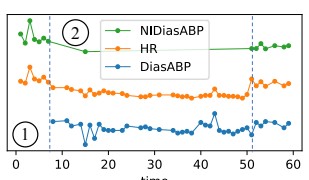

Figure 1: Illustration of three variables in an ISMTS sample. In the first 7 seconds (Box 1), a strong correlation between HR and NIDiasABP is observed. As NIDiasABP becomes more sparse, the correlation between HR and NIDiasABP weakens between 7 and 50 seconds (Box 2), while the correlation between HR and DiasABP increases.

To this end, we aim to explicitly consider the differences and correlations among variables in ISMTS, empowering the model to capture fine-grained features at the variable level. However, the sampling rates, sampling times, observation spans, and observation lengths of different variables vary in ISMTS, as shown in three subgraphs in Figure 2. This makes it tough and complex to infer these two aspects only from the time series modality of variables. We rethink the issue based on an intuitive observation as shown in Figure 2: Variables that exhibit (dis)similar temporal patterns frequently have (dis)similar medical properties in reality, which motivates us to consider the differences and correlations of variables from the perspective of domain knowledge directly. Recent work [11, 12] has successfully represented domain knowledge through textual modal information, enhancing model performance in medical imaging, which provides us with insights. Specifically, the medical properties of each variable can be described in natural language. Leveraging the powerful semantic understanding capabilities of the Pre-trained Language model (PLM), we can obtain semantic representations from the textual knowledge of each variable. This set of textual representations forms an overall view of variables from the perspective of medical knowledge, clearly showing inherent differences and correlations among variables—exactly what we need.

Based on the above analysis, we propose the Knowledge-Empowered Dynamic Graph Network (KEDGN), which utilizes textual semantic representations of variables obtained through PLM as guidance. On this basis, we 1) allocate unique parameter space for each variable to capture their specific temporal pattern, 2) generate a complete variable graph and introduce a density-aware mechanism to explicitly model time-varying correlations among variables in ISMTS. Finally, these

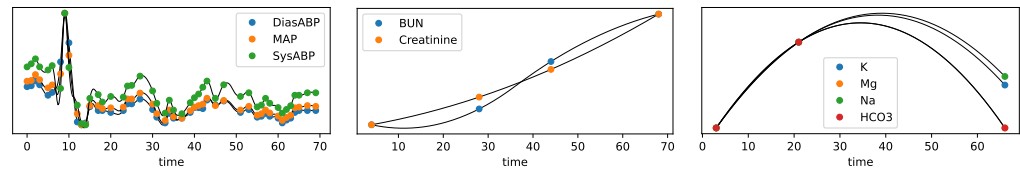

(a) Invasive arterial blood pressure  (b) Renal function indicators  (c) Ion concentration indicators

Figure 2: The time patterns (sampling rates, sampling times, observation spans, observation lengths, trends, etc.) of variables among different subgraphs exhibit significant differences, as they have distinct medical properties. Meanwhile, variables within the same subgraph share similar time patterns, and their medical properties are closely related. (More variable groups can be found in Figure 7).

two modules are integrated with a graph convolutional recurrent network to capture both temporal and inter-variable dependencies in ISMTS. Our contributions can be summarized in three aspects:

- We leverage variable-specific textual medical knowledge to empower the model to capture variable-specific temporal patterns in ISMTS distinctively.

- We introduce a density-aware mechanism based on the knowledge-empowered variable graph to model the time-varying inter-variable dependencies in ISMTS.

- Empirical results on four real-world medical datasets demonstrate that KEDGN outperforms state-of-the-art methods. Visualization analysis further illustrates the strong interpretability of our approach.

## 2   Related Work

**Irregularly Sampled Multivariate Time Series Modeling**   Existing methods can be roughly categorized into interpolation-based and raw data-based approaches. The former, employing methods such as kernel-based approaches [13, 14], Gaussian process [15] or hourly aggregation [16], aims to obtain a set of regularly spaced observations. However, interpolation may result in the loss of useful information about the original sequences, such as missing patterns. The latter, raw data-based methods, directly learn from irregular time series. To adapt to uneven sampling intervals, [17] improves recurrent neural networks, [3, 4] introduce neural ordinary differential equations and [6, 5, 18, 19] adopt time embeddings. [20] converts ISMTS into line graph images and utilizes pretrained vision transformers for extracting features. These methods primarily focus on overall temporal dependencies, needing more consideration for fine-grained variable-level patterns and correlations. Despite recent research introducing attention [21, 22] or graph neural networks [9, 23] to account for variable correlations, these methods have limited performance due to the lack of prior knowledge and the use of shared parameter spaces among all variables.

**Graph Neural Networks for Multivariate Time Series**   In recent years, a series of studies have integrated GNN with various time series modeling frameworks to effectively capture both inter-variable and inter-temporal dependencies in MTS [24]. These approaches have been widely applied in diverse domains, including transportation [25], healthcare [26], economics [27], demonstrating promising results in mainstream tasks such as prediction [28], classification [29], and imputation [30]. Although recent work [31, 32, 33, 34] has proposed the idea of modeling variable relationships through learning dynamic graphs, most of these methods are primarily designed for regularly sampled MTS with synchronous observations, and further improvements are needed to adapt them for irregularly sampled time series.

**Medical Knowledge Enhanced Models**   Several studies have utilized the rich domain knowledge in the medical field to enhance models. [12, 35] apply knowledge for computing additional features, while [36, 37] utilize knowledge to guide the final training loss, demonstrating the effectiveness of medical prior knowledge. However, existing methods commonly focus on visual language pretraining in medical scenes or medical report generation [11]. How to effectively integrate domain knowledge to guide medical time series modeling remains a challenge.

## 3   Problem Definition

Given a dataset $\mathcal{D} = \{(s_i, \ y_i) \mid i = 1, \ldots, N\}$ containing $N$ patient samples, the $i$-th sample consists of an irregular multivariate time series $s_i$ and a label $y_i$. For the dataset with a total variable count of $V$ and a maximum sample observation length of $T$, $s_i$ can be denoted as a tuple: $s_i = (t_i, x_i, m_i)$, where $t_i \in \mathbb{R}^T$ represents the observation timestamps, $x_i \in \mathbb{R}^{V \times T}$ represents the multivariate time series observations, and any unobserved values or the missing parts of the time series shorter than the maximum sample observation length are filled with 0. The binary indicator $m_i$ has the same size as $x_i$, indicating which elements in the $x_i$ are actually observed. We use 1 to represent observed values and 0 to represent missing values. In this paper, we focus on patient mortality and morbidity prediction, i.e., classification task, aiming to correctly predict class label $\hat{y}_i$ given a sample $s_i$.

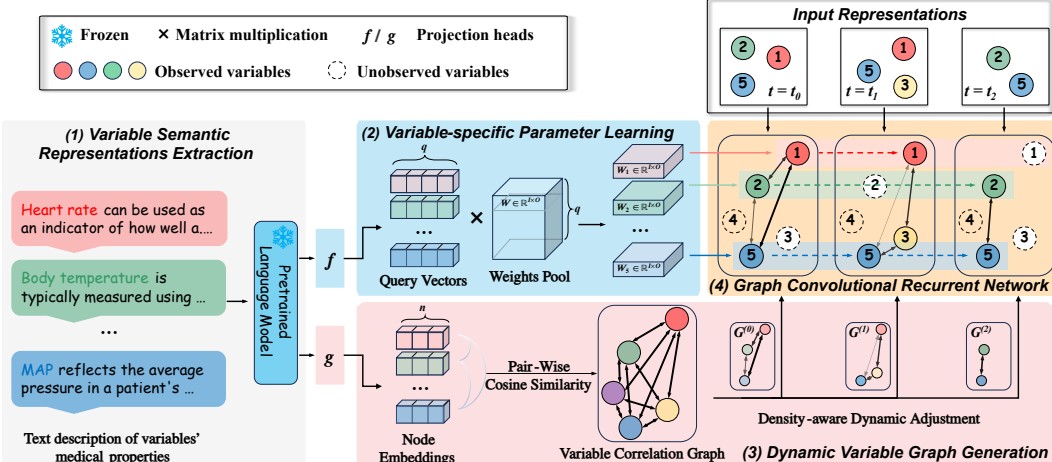

Figure 3: The model framework of KEDGN. We (1) utilize a PLM to extract semantic representations for each variable from textual medical properties (Section 4.1). Based on this, we (2) allocate variable-specific parameter space to capture variable-specific temporal patterns (Section 4.2), (3) generate dynamic variable graphs by combining knowledge-empowered graph with a density-aware mechanism to model time-varying correlations among variables (Section 4.3). (4) The above two modules are injected into graph convolutional recurrent network to model intra-variable and inter-variable dependencies in ISMTS simultaneously (Section 4.4).

## 4 The Proposed Model

### 4.1 Variable Semantic Representations Extraction

First, we introduce how to extract semantic representation for each variable from medical knowledge. Let $\mathcal{V} = \{v_1, v_2, \ldots, v_V\}$ be the set of variables, and a descriptive sentence of medical properties associated with the $j^{th}$ variable can be denoted as:

$$P_j = \left\{w_{j,1}, w_{j,2}, ..., w_{j,l_j} \,\middle|\, j = 1, 2, \ldots, V\right\}, \tag{1}$$

where $l_j$ is the length of the $j^{th}$ variable's sentence and $w_{j,i}$ denotes the $i^{th}$ word of the $j^{th}$ sentence. We leverage a PLM to represent each text description $P_j$ as a $d$-dimensional embedding. Considering the diverse types of PLMs with varying methods of utilization, we use the widely adopted encoder-based model BERT [38] as an illustration:

$$e_j = \mathbf{BERT}([\text{CLS}], w_{j,1}, w_{j,2}, ..., w_{j,l_j}, [\text{SEP}]) \in \mathbb{R}^{(l_j+2) \times d}, \tag{2}$$

where [CLS] and [SEP] are special tokens indicating a sequence's beginning and end, respectively. BERT generates an embedding for each token of the input sequence. Since the embedding at the [CLS] position captures the semantic information of the entire input sequence, we take the hidden state at the [CLS] position as the overall semantic representation of each variable: $E_j = e_j([\text{CLS}])$. This yields a semantic representation matrix $\boldsymbol{E} = [E_1, E_2, ..., E_V] \in \mathbb{R}^{V \times d}$ and forms an overall view of variables from the perspective of medical knowledge.

### 4.2 Variable-specific Parameter Learning

Since temporal patterns of ISMTS vary from variable to variable, simply using shared parameter space for all variables is insufficient to capture differentiated temporal dependencies. In this section, we adjust parameter space for different variables to adapt to differentiated temporal patterns based on the extracted variables' semantic representations. For any parameter matrix $w \in \mathbb{R}^{I \times O}$ with input dimension $I$ and output dimension $O$, the total parameter space needed for $V$ variables is $\Theta \in \mathbb{R}^{V \times I \times O}$. Inspired by [39], we decompose $\Theta$ into two matrices: a variable representation matrix $\boldsymbol{Q} \in \mathbb{R}^{V \times q}$ and a weight pool matrix $\boldsymbol{W} \in \mathbb{R}^{q \times I \times O}$, where $q$ is a hyperparameter for the intermediate dimension. Here, $\boldsymbol{Q}$ consists of $q$-dimensional query vectors for $V$ variables used to distinguish differences among variables and obtain variable-specific parameters from the weight pool.

We use a projection $f(\cdot) : \mathbb{R}^d \rightarrow \mathbb{R}^q$ to variable semantic representations $\boldsymbol{E}$ for obtaining query vectors rather than directly using $\boldsymbol{E}$. On the one hand, the dimension of the query vector $q$ directly determines the size of the weight pool, but the output dimension of PLM is often large (e.g., 768 in BERT), leading to a sharp increase in model complexity. On the other hand, there is a modality gap between textual embeddings and the temporal parameter space. The projection $f(\cdot)$ can achieve both feature reduction and modality transformation. In our implementation, we use a nonlinear projection with one additional hidden layer (and ReLU activation). Thus, the parameter space specific to variable $i$ can be obtained using the following formula:

$$\boldsymbol{\Theta_i} = f(E_i)\boldsymbol{W} \in \mathbb{R}^{I \times O}, \tag{3}$$

The approach we employ to generate variable-specific parameter space is general and not restricted to a specific model backbone because any model architecture is composed of multiple parameter matrices $W$.

### 4.3 Dynamic Variable Graph Generation

In this section, we introduce how to generate dynamic correlation graphs of variables for explicitly modeling the time-varying correlations among variables in ISMTS.

#### 4.3.1 Complete Variable Correlation Graph Learning

The misaligned and imbalanced observations of the variables in ISMTS make it difficult to learn the variable correlations from the time series. Therefore, we extract a static complete variable correlation graph based on the textual semantic representations of the variables directly from the perspective of the actual medical properties of variables. We apply another non-linear projection $g(\cdot) : \mathbb{R}^d \rightarrow \mathbb{R}^n$ to the textual representations of variables $\boldsymbol{E}$ to obtain $n$-dimensional node embeddings for each variable. Subsequently, we calculate the pairwise cosine similarity among the node embeddings of variables, resulting in a $V \times V$ matrix of variable similarity. Finally, we use the softmax function to normalize the edge weights corresponding to each node, producing a normalized graph of variable correlations. The correlation weight between the $i^{th}$ and $j^{th}$ variables can be calculated as:

$$A_{ij} = \text{Softmax}(\frac{g(E_i) \cdot g(E_j)}{\|g(E_i)\| \cdot \|g(E_j)\|}), \tag{4}$$

where $\cdot$ represents vector dot product and $\|\cdot\|$ represents the vector magnitude. The introduction of $g(\cdot)$ in this context not only performs feature reduction but also avoids using a completely fixed prior graph. It preserves the model's ability to adaptively optimize the graph structure based on different data distributions and downstream tasks. Thus, we obtain a knowledge-empowered complete graph with $V$ nodes to measure the static correlation among variables in general medical cases, and its adjacency matrix is denoted as $\boldsymbol{A}$.

#### 4.3.2 Dynamic Density-aware Adjustment Mechanism

Due to the varying subsets of variables observed at each timestamp in ISMTS, we use different subgraphs of $\boldsymbol{A}$ to describe the variable correlations at different timestamps. Specifically, we use a mask matrix $\boldsymbol{M}^{(t)} \in \mathbb{R}^{V \times V}$ to indicate the subgraph topology at timestamp $t$:

$$\boldsymbol{M}_{ij}^{(t)} = \begin{cases} 1, & \text{if both variables } i \text{ and } j \text{ are observed at } t \\ 0, & \text{otherwise} \end{cases}, \tag{5}$$

Therefore, we can calculate the variable correlation subgraph $\boldsymbol{A}^{(t)}$ at timestamp $t$ through $\boldsymbol{A}^{(t)} = \boldsymbol{A} \odot \boldsymbol{M}^{(t)}$, where $\odot$ represents Hadamard product. Additionally, we introduce a density-aware mechanism to dynamically adjust edge weights of subgraphs in different timestamps to fit in the time-varying correlations among variables mentioned in Figure 1. Specifically, we estimate the sampling density of any observation point by considering the average time interval between each observation point and its preceding and succeeding observations. If there is no preceding/succeeding observation, we take the time interval of the succeeding/preceding observation as the density. If neither a preceding nor a succeeding observation exists, it indicates that this observation is the only

one for the variable, and we take half of the maximum observation time span as the density. The formula for calculating the sampling density of the $i$-th observation of variable $v$ at timestamp $t$ is:

$$Z^{(t)} = Z_{i,v} = \begin{cases} ((t_{i,v} - t_{i-1,v}) + (t_{i+1,v} - t_{i,v}))/2, \text{if both } t_{i+1,v} \text{ and } t_{i-1,v} \text{ exist} \\ t_{i,v} - t_{i-1,v}, \quad \text{if } t_{i+1,v} \text{ does not exist} \\ t_{i+1,v} - t_{i,v}, \quad \text{if } t_{i-1,v} \text{ does not exist} \\ t_{max}/2, \quad \text{if neither } t_{i+1,v} \text{ nor } t_{i-1,v} \text{ exists.} \end{cases} \tag{6}$$

Then we calculate the density scores for various variables at timestamp $t$ through:

$$D^{(t)} = \alpha\sigma(Z^{(t)}) \in \mathbb{R}^V, \tag{7}$$

where $\sigma$ is an activation function and $\alpha$ is a hyperparameter that controls the proportion. At timestamp $t$, the edge weight between the $i^{th}$ and $j^{th}$ variables is adjusted as:

$$\boldsymbol{G}^{(t)}_{ij} = \boldsymbol{A}^{(t)}_{ij} \times (1 - W_{ij}|D^{(t)}_i - D^{(t)}_j|), \tag{8}$$

where $W \in \mathbb{R}^{V \times V}$ is a learnable parameter matrix. Thereby, we achieve the dynamic adjustment of the variable graph weights in response to changes in variable sampling density.

## 4.4 Variable-specfic Dynamic Graph Convolutional Recurrent Network

Under the empowerment of variable textual representation, we have obtained variable-specific parameters $\boldsymbol{\Theta} \in \mathbb{R}^{V \times I \times O}$ and dynamic variable graph $\boldsymbol{G} \in \mathbb{R}^{T \times V \times V}$. In this section, we integrate these two modules into the graph convolutional neural network to handle ISMTS. GCRNN [40] is a backbone network that introduces graph convolutional operations on top of an RNN variant, Gated Recurrent Unit [41]. This structure is simple, effective, and easy-to-adapt for ISMTS, as it enables variable-level parallel computation of asynchronous observations without explicit interpolation. Specifically, we allocate a unique hidden state for each variable, updating the state only at the observed timestamps to avoid imputation and preserve the individual sampling patterns of each variable. The graph convolution operation over a graph signal $S \in \mathbb{R}^{V \times I}$ containing $V$ nodes at timestamp $t$ is defined as follows:

$$\boldsymbol{\Theta} \star_{G^{(t)}} S \approx (I_V + G^{(t)})^T S \times \boldsymbol{\Theta}, \tag{9}$$

where $I_V \in \mathbb{R}^{V \times V}$ is identity matrix, $\times$ represents batch matrix multiplication. Here, we adopt $1^{st}$-order Chebyshev polynomial expansion approximation [42] for graph convolution. The updated formulas for variable states at timestamp $t$ are:

$$r^{(t)} = \sigma(\Theta_r \star_{G^{(t)}} [X^{(t)}||H^{(t-1)}] + b_r), \tag{10}$$

$$u^{(t)} = \sigma(\Theta_u \star_{G^{(t)}} [X^{(t)}||H^{(t-1)}] + b_u), \tag{11}$$

$$C^{(t)} = tanh(\Theta_C \star_{G^{(t)}} [X^{(t)}||(r^{(t)} \odot H^{(t-1)})] + b_C), \tag{12}$$

$$H^{(t)}_i = \begin{cases} H^{(t-1)}_i, \quad \text{if variable } i \text{ is unobserved at time } t \\ u^{(t)}_i \odot H^{(t-1)}_i + (1 - u^{(t)}_i) \odot C^{(t)}_i, \text{otherwise} \end{cases}, \tag{13}$$

where $||$ denotes the concatenate operation, $H^{(t-1)} \in \mathbb{R}^{V \times h}$ is the variable states at the previous timestamp and $X^{(t)} \in \mathbb{R}^{V \times k}$ denotes the input representation at current timestamp. We follow the structured input encoding method of [22], using multiple fully-connected mappings to encapsulate each observed value and its corresponding timestamp into a $k$-dimensional input representation (All 0 vectors for unobserved values) to indicate flexible observation time and adapt to the uneven intervals within variables. $r^{(t)}, u^{(t)} \in \mathbb{R}^{V \times h}$ are reset gate and updated gate, respectively. $\Theta_r, \Theta_u, \Theta_C \in \mathbb{R}^{V \times (k+h) \times h}$ are variable-specific parameters obtained by respectively multiplying the query vectors matrix $\boldsymbol{Q} \in \mathbb{R}^{V \times q}$ with three weight matrices $W_r, W_u, W_C \in \mathbb{R}^{q \times (k+h) \times h}$.

We calculate the sum of $h$ channels for each variable's hidden state $H_i$ at the last observed timestamp to get a $V$-dimensional vector $C$. Additionally, we follow the approach used in [9] to incorporate the static features. Specifically, static features of each sample are mapped into a static vector $S$ through a linear layer. Finally, $C$ and $S$ are concatenated to predict the final classification probabilities: $\hat{y} = \text{Softmax}(W^y[C||S] + b^y)$. The training objective is minimizing the cross-entropy loss between $\hat{y}$ and $y$. The pseudo-code for KEDGN is presented in Appendix A (Algorithm 1).

# 5 Experiment

## 5.1 Experimental Setting

**Datasets and Baselines**   We conduct experiments on four widely used irregular medical time series datasets, namely P19 [43], Physionet [44], MIMIC-III [45] and P12 [46] where Physionet is a reduced version of P12 considered by prior work [6]. We compare our method with the state-of-the-art methods for modeling irregular time series, including GRU-D [17], ODE-RNN [4], IP-Net [14], SeFT [5], mTAND [6], Raindrop [9], StraTS [18], DuETT [19], ViTST [20] and Warpformer [22]. In addition, we also compare our method with two approaches initially designed for forecasting tasks, namely $DGM^2$-O [13] and MTGNN [47]. The implementation and hyperparameter settings of these baselines are kept consistent with those used in [9]. More details of datasets and baselines can be found in Appendix B and C.

**Evaluation Setup**   For the data pre-processing of MIMIC-III, we follow the method described in [48] and divide the dataset into three parts for training, validation, and testing with the ratio of 70%,15%,15%. For the remaining three datasets, we adhered to [9] 's approaches, and the ratio of training, validation, and testing set is 8:1:1. We measure the classification performance with the Area Under the Receiver Operating Characteristic Curve (AUROC) and Area Under the Precision-Recall Curve (AUPRC) since all the four datasets are binary classification datasets with highly imbalanced class distribution. AUPRC has better sensitivity to sample imbalance [49]; thus, the optimal model parameters that achieve the best AUPRC on the validation set are used for the test set. More details of metrics can be found in Appendix D.

**Implementation Details**   We adopt the Adam [50] optimizer, and the number of training epochs is set as 10. Due to differences in dataset sizes, the learning rate is set as 0.001 for Physionet and P12 and 0.005 for MIMIC-III and P19. The textual sources for variable descriptions are flexible; we chose three sources, including the variable's full name, Wikipedia source, and ChatGPT source (Default), corresponding to model names KEDGN-Name, KEDGN-Wiki, and KEDGN-ChatGPT, respectively. All experiments are conducted with five random seeds, and the average and standard deviation are reported. More implementation details and hyperparameter settings can be found in Appendix E.

## 5.2 Main Results

Table 1: Method benchmarking on irregularly sampled medical time series classification. The best results are highlighted in **bold**, and the second-best results are in underlined. The results in the table are presented in the form of (Mean ± Std %).

| Methods | P19 | | Physionet | | MIMIC-III | | P12 | |
|---|---|---|---|---|---|---|---|---|
| | AUROC | AUPRC | AUROC | AUPRC | AUROC | AUPRC | AUROC | AUPRC |
| GRU-D | 88.7 ± 1.2 | 57.6 ± 2.3 | 79.1 ± 6.9 | 42.7 ± 7.2 | 82.2 ± 1.8 | 43.3 ± 2.1 | 79.6 ± 0.6 | 41.7 ± 1.8 |
| ODE-RNN | 87.1 ± 1.0 | 52.6 ± 3.2 | 75.5 ± 2.8 | 33.7 ± 4.1 | 81.0 ± 0.6 | 42.3 ± 0.7 | 78.8 ± 0.6 | 37.4 ± 2.6 |
| IP-Net | 90.2 ± 0.2 | 58.6 ± 0.8 | 86.8 ± 0.6 | 55.8 ± 1.4 | 84.1 ± 0.1 | 47.1 ± 0.9 | 83.7 ± 0.3 | 46.3 ± 1.3 |
| SeFT | 84.0 ± 0.3 | 49.3 ± 0.5 | 75.5 ± 0.2 | 29.4 ± 0.9 | 67.9 ± 0.2 | 23.2 ± 0.4 | 78.1 ± 0.5 | 35.9 ± 0.8 |
| MTGNN | 88.5 ± 1.0 | 55.8 ± 1.5 | 77.1 ± 4.4 | 35.4 ± 7.3 | 78.5 ± 2.3 | 35.2 ± 3.1 | 82.1 ± 1.5 | 41.8 ± 2.1 |
| mTAND | 82.9 ± 0.9 | 32.2 ± 1.5 | 86.9 ± 1.3 | 52.5 ± 1.3 | 83.8 ± 0.3 | 46.6 ± 0.5 | 85.3 ± 0.3 | 49.3 ± 1.0 |
| $DGM^2$-O | 91.6 ± 0.5 | 60.0 ± 1.3 | 85.8 ± 0.7 | 50.4 ± 3.2 | 81.0 ± 0.9 | 37.6 ± 1.1 | 85.8 ± 0.1 | 48.3 ± 0.7 |
| Raindrop | 89.4 ± 0.6 | 61.2 ± 1.1 | 82.7 ± 1.4 | 41.2 ± 3.6 | 79.8 ± 1.3 | 35.2 ± 1.1 | 82.2 ± 1.1 | 43.3 ± 2.1 |
| StraTS | 91.2 ± 0.3 | 58.4 ± 1.4 | 84.9 ± 1.5 | 47.3 ± 5.3 | 84.4 ± 0.4 | 46.4 ± 0.8 | 86.7 ± 0.7 | 52.1 ± 1.5 |
| DuETT | 88.2 ± 0.5 | 56.0 ± 3.9 | 81.3 ± 1.4 | 44.9 ± 1.4 | 78.8 ± 0.8 | 34.3 ± 1.0 | 83.4 ± 1.2 | 45.4 ± 1.5 |
| ViTST | 91.7 ± 0.1 | 57.5 ± 0.7 | 81.3 ± 1.9 | 37.4 ± 2.9 | 81.8 ± 0.3 | 39.6 ± 1.3 | 86.3 ± 0.1 | 50.8 ± 1.5 |
| Warpformer | 91.8 ± 0.4 | 60.6 ± 2.6 | 83.3 ± 0.7 | 43.5 ± 2.3 | 84.6 ± 0.5 | 47.4 ± 0.9 | 85.4 ± 0.5 | 50.4 ± 1.5 |
| **KEDGN-Name** | **92.3 ± 1.0** | **62.5 ± 0.7** | 87.9 ± 1.4 | 56.0 ± 3.2 | 84.8 ± 0.3 | **48.4 ± 1.5** | 87.1 ± 0.8 | 54.1 ± 2.6 |
| **KEDGN-Wiki** | 92.2 ± 0.6 | 62.3 ± 1.4 | **88.2 ± 1.1** | **57.5 ± 2.5** | 84.3 ± 0.9 | 47.7 ± 1.8 | 87.0 ± 0.3 | 53.1 ± 0.5 |
| **KEDGN-ChatGPT** | 92.2 ± 0.5 | 62.0 ± 1.3 | 87.9 ± 0.2 | 57.1 ± 1.8 | **85.1 ± 0.3** | 48.3 ± 1.6 | **87.8 ± 0.5** | **54.5 ± 1.5** |

**Classic time series classification**   The evaluation results are summarized in Table 1, in which we use BERT to extract variables' semantic representations. More experimental results and analyses using other PLMs can be found in Appendix F.1. Overall, our KEDGN achieves the best performance on all four datasets and outperforms the strongest baseline by an average of 0.9% on AUROC and

1.6% on AUPRC. In addition, the results of models using different text sources are similar, which demonstrates that our method is not limited to a specific text source and exhibits generalizability. We also provide an analysis of computational costs in Appendix F.2.

**Leave-variables-out**   To demonstrate the robustness of our method, we test whether KEDGN can achieve good performance when a subset of variables is completely missing. We uniformly discard 10%, 20%, 30%, 40%, and 50% of the variables, hiding all their observations in both validation and test sets. Table 2 reports the results on the MIMIC-III dataset, while the results for the remaining datasets are presented in Appendix F.3 (Table 8). Our method achieves the highest performance on 35 out of 40 metrics across 4 datasets as the missing rate increases from 10% to 50%. This may be attributed to KEDGN only handling actually observed points, avoiding the accumulation of imputation errors, particularly in cases of higher missing ratios, thus exhibiting a degree of robustness.

Table 2: Classification performance on samples with a fixed set of left-out variables on the MIMIC-III dataset. The best results are highlighted in **bold** and the second best results are in underlined.

| Methods | Discard ratio | | | | | | | | | |
|---|---|---|---|---|---|---|---|---|---|---|
| | 10% | | 20% | | 30% | | 40% | | 50% | |
| | AUROC | AUPRC | AUROC | AUPRC | AUROC | AUPRC | AUROC | AUPRC | AUROC | AUPRC |
| GRU-D | 81.0 ± 0.6 | 42.1 ± 0.8 | 80.3 ± 0.9 | 41.7 ± 1.0 | 79.2 ± 1.8 | 41.0 ± 1.4 | 78.5 ± 2.1 | 40.4 ± 1.6 | 77.9 ± 2.2 | 39.9 ± 1.8 |
| mTAND | 81.2 ± 0.2 | 42.1 ± 0.8 | 80.4 ± 1.1 | 41.9 ± 1.2 | 79.7 ± 1.4 | 41.0 ± 1.7 | 79.3 ± 1.4 | 40.4 ± 2.0 | 78.8 ± 1.6 | 39.8 ± 2.3 |
| DGM²-O | 78.8 ± 0.5 | 34.2 ± 0.9 | 78.3 ± 0.8 | 33.9 ± 1.1 | 77.6 ± 1.2 | 33.4 ± 1.2 | 77.3 ± 1.3 | 33.1 ± 1.2 | 76.8 ± 1.5 | 32.6 ± 1.4 |
| MTGNN | 78.8 ± 1.1 | 34.5 ± 1.4 | 78.0 ± 1.6 | 34.0 ± 1.3 | 77.1 ± 2.2 | 33.5 ± 1.5 | 76.3 ± 2.5 | 32.8 ± 1.9 | 75.6 ± 3.2 | 32.2 ± 2.4 |
| Raindrop | 78.2 ± 1.1 | 33.7 ± 0.9 | 77.5 ± 1.3 | 33.5 ± 0.9 | 76.4 ± 2.1 | 32.8 ± 1.4 | 76.0 ± 2.0 | 32.5 ± 1.4 | 75.7 ± 2.0 | 32.3 ± 1.4 |
| DuETT | 78.0 ± 0.5 | 34.0 ± 0.9 | 77.2 ± 1.0 | 33.7 ± 0.8 | 76.6 ± 1.2 | 33.3 ± 1.0 | 76.4 ± 1.2 | 33.0 ± 1.0 | 76.1 ± 1.3 | 32.6 ± 1.3 |
| Warpformer | 82.5 ± 0.5 | 43.1 ± 0.8 | 81.7 ± 0.9 | 42.5 ± 1.2 | 81.2 ± 1.1 | 42.1 ± 1.2 | 80.6 ± 1.5 | 41.8 ± 1.3 | 80.0 ± 1.9 | 41.3 ± 1.6 |
| **KEDGN** | **83.0 ± 0.7** | **44.8 ± 2.0** | **82.3 ± 1.1** | **44.4 ± 1.9** | **81.3 ± 1.9** | **43.6 ± 2.2** | **80.6 ± 2.1** | **43.0 ± 2.4** | **80.0 ± 2.3** | **42.4 ± 2.5** |

## 5.3   Ablation Study

In this section, we investigate the performance benefits generated by each key component of the proposed method on all four datasets. We compare the full versioned method with its six variants: (1) **w/o VSW**: We apply shared RNN parameter weights for all variables; (2) **w/o Text**: We replace the variable-specific textual semantic representations $E_i$ with randomly initialized learnable embeddings; (3) **w/o Graph**: We set the graph $G$ to be a fully zero matrix, disregarding dependencies among variables; (4) **w/o KEE**: We replace knowledge-empowered node embeddings of variables $g(E_i)$ with randomly initialized learnable embeddings, (5) **w/o DAG**: We remove the density-aware adjustment for edge weights of the graph, using static adjacency matrix $A^{(t)}$ during different periods. (6) **w/o TE**: We remove the timestamp embedding part of the structured input encoding $X^{(t)}$. The results on the P19 dataset are presented in Table 3, while the results for the

Table 3: The ablation study of our proposed method KEDGN on P19. The results in the table are presented in the form of (Mean ± Std %).

| Metrics | AUROC | AUPRC |
|---|---|---|
| w/o VSW | 91.5 ± 0.3 | 56.9 ± 0.4 |
| w/o Text | 91.7 ± 0.6 | 60.6 ± 3.0 |
| w/o Graph | 90.8 ± 0.8 | 58.3 ± 1.6 |
| w/o KEE | 91.5 ± 0.6 | 60.4 ± 2.1 |
| w/o DAG | 91.6 ± 0.4 | 60.0 ± 1.3 |
| w/o TE | 91.4 ± 0.5 | 58.1 ± 2.6 |
| **Full** | **92.2 ± 0.5** | **62.0 ± 1.3** |

remaining datasets are presented in Appendix F.4 (Table 9). The results show that all model components are necessary and variable-specific parameter space makes the most significant contribution to the performance of KEDGN.

## 5.4   Visualization Analysis

### 5.4.1   Visualization of Variables Textual Representations

In this section, we explore why textual information is effective for time series modeling through visualization analysis. We first group variables with similar time patterns on the P12 dataset, as illustrated in Figure 2 (More variables groups can be found in Appendix F.7 (Figure 7)). Subsequently, we use T-SNE [51] to visualize the distribution of variable semantic representations. Figures 4a and 4b respectively display the distributions of ChatGPT and Wikipedia sources, while Figure 4c shows the final distribution learned by replacing the text representation with randomly initialized learnable embeddings. Variables with similar time patterns are labeled with the same color. It can be observed

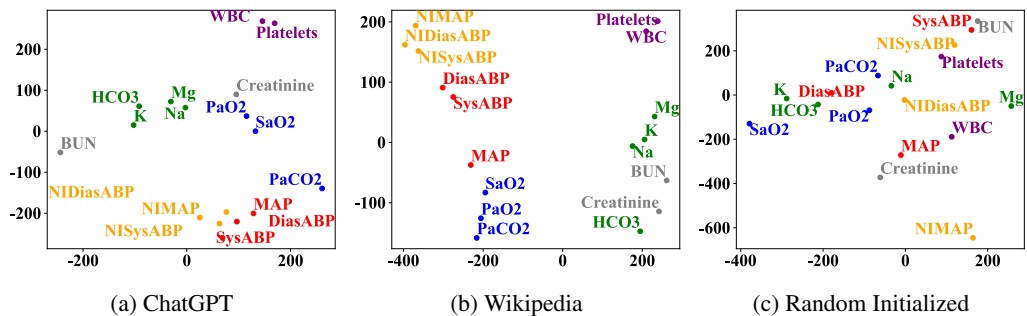

(a) ChatGPT        (b) Wikipedia        (c) Random Initialized

Figure 4: T-SNE visualization of partial variable representations on the P12 dataset.

that the textual representation space of variables exhibits distinct clustering. Although there may be occasional outliers when using different text sources, such as BUN in Figure 4a and HCO3 in Figure 4b, the overall clusters are generally consistent with the variable groups we divide based on time series patterns. However, the learnable variable embeddings in Figure 4c, optimized by classification loss, tend to be distributed uniformly, which is difficult to effectively reflect the intrinsic differences among variables. More visualizations of other datasets and other PLMs can be found in Appendix F.8 and F.9, respectively. Based on this phenomenon, we infer that text descriptions and time series are both external manifestations of the inherent sense of variables; they just belong to different data modalities. These two forms of data for the same variable should exhibit relative consistency. Therefore, the relative distribution among variables extracted from text and time series should ideally be similar. Leveraging PLM allows for the straightforward and efficient extraction of this universal view from textual descriptions, which is equally applicable to describing the relative distribution of temporal patterns among variables. Thus, the effectiveness of PLM and the cross-modal relative consistency are the keys to guiding time series modeling based on textual information.

### 5.4.2 Visualization of Variable Correlation Graph

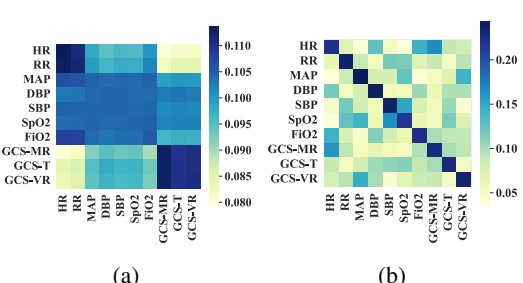

(a)            (b)

Figure 5: Visualization of the learned correlation graph of variables on the MIMIC-III dataset.

We visualize the learned partial inter-variable correlation graphs on the MIMIC-III dataset in the form of heatmaps. In Figure 5a, we depict the graph learned based on node embeddings mapped from textual representations, while in Figure 5b, the graph is learned based on randomly initialized learnable node embeddings. We observe that the top-left and bottom-right corners of the heatmap in Figure 5a exhibit darker colors, indicating strong correlations consistent with medical domain knowledge. Specifically, the variables in the top-left corner, Heart Rate (HR) and Respiration Rate (RR), are commonly monitored together in clinical settings for assessing vital signs and respiratory system function [52]. The variables in the bottom-right corner, GCS-MR, GCS-T, and GCS-VR, collectively constitute the Glasgow Coma Scale (GCS), which is used to assess a patient's neurological status and level of consciousness [53]. These correlations are not evident in the graph without textual representations. This once again validates our perspective that relying solely on downstream task optimization for adaptive learning in graphs in ISMTS is insufficient to reflect the actual medical correlations among variables and lacks interpretability. In contrast, textual representations can guide the model to accurately extract variable correlations aligned with domain knowledge to provide high interpretability.

### 5.4.3 Visualization of Dynamic Density-aware Graph

In Figure 6, we visualize the dynamic density-aware graph for the sample in Figure 1. We present the time series of three variables and the corresponding correlation heatmaps learned by our model at

timestamps $4$, $15$ and $56$. From the time series, we observe that around $t = 4$, HR shows a strong correlation with NIDiasABP, while the correlation with DiasABP is masked as 0 since DiasABP has not been observed yet. Around $t = 15$, the correlation between HR and NIDiasABP decreases, while a relatively strong correlation with DiasABP emerges. By $t = 56$, HR exhibits a strong correlation with both variables. This process is clearly reflected in the heatmaps: the color between HR and NIDiasABP transitions from dark to light from $t = 4$ to $t = 15$ and darkens again from $t = 15$ to $t = 56$. The color between HR and DiasABP remains dark at $t = 15$ and $t = 56$. This demonstrates that our dynamic density-aware mechanism exactly reflects the time-varying correlations among variables in ISMTS.

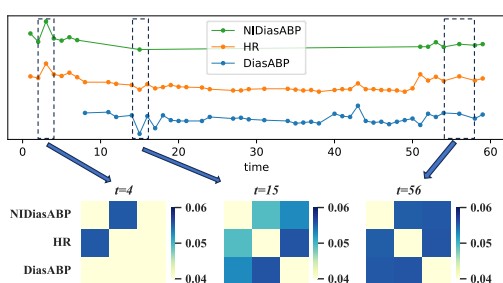

Figure 6: Visualization of the dynamic graph of three variables over time for the sample in Figure 1. To enhance the contrast ratio, we set the diagonal elements to 0.

## 6  Limitations

Although our proposed method effectively guides ISMTS modeling through the domain knowledge from text modality, it has some limitations. The backbone of our model is a recurrent-based architecture, which inherently has a sequential computation characteristic that can be a bottleneck in terms of runtime. Additionally, our method is specifically tailored for medical applications, and its performance may be limited in other irregular multivariate time series applications, such as human activity recognition, where variables lack domain knowledge and thus cannot generate high-quality text descriptions.

## 7  Conclusion

In this paper, we propose KEDGN for modeling ISMTS. The proposed method leverages a PLM to flexibly extract semantic representation for each variable from the textual medical knowledge. Based on these representations, we allocate the variable-specific parameter space to capture variable-specific temporal patterns and extract a complete variable graph as a measure of the variables' static medical correlations. Considering the time-varying variables correlations in ISMTS, we introduce a density-aware mechanism to dynamically adjust the subgraph across different periods. Our experimental results demonstrate that KEDGN outperforms existing methods in ISMTS classification tasks and provides high interpretability. Our future work will focus on investigating the applicability of KEDGN in a range of related tasks, such as interpolation, extrapolation, and regression.

## Acknowledgements

The work described in this paper was partially funded by the National Natural Science Foundation of China (Grant No. 62272173), the Natural Science Foundation of Guangdong Province (Grant Nos. 2024A1515010089, 2022A1515010179), the Science and Technology Planning Project of Guangdong Province (Grant No. 2023A0505050106), and the National Key R&D Program of China (Grant No. 2023YFA1011601). Yicheng Luo and Zhen Liu equally contributed to this work.

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

# A  Algorithm of KEDGN

---

**Algorithm 1** The pseudo-code of KEDGN

---

**Input**: An ISMTS sample observation values matrix: $x^{(1:T)} \in \mathbb{R}^{V \times T}$, binary mask matrix $m^{(1:T)} \in \mathbb{R}^{V \times T}$, observation timestamps $t^{(1:T)} \in \mathbb{R}^T$, the sentences of $V$ variables' medical properties $\mathcal{P} = \{P_1, P_2, \ldots, P_V\}$

**Output**: Predicted classification label $\hat{y}$

1: extract semantic representations $\boldsymbol{E} \in \mathbb{R}^{V \times d}$ for variables from $\mathcal{P}$ through PLM using Eq.(2)
2: obtain variable-specific parameter space $\boldsymbol{\Theta}$ (including reset gate $\Theta_r$, update gate $\Theta_u$ and candidate gate $\Theta_C \in \mathbb{R}^{V \times (k+h) \times h}$) using Eq.(3)
3: obtain complete variable correlation graph $\boldsymbol{A}$ using Eq.(4)
4: initialize the hidden state of $V$ variables $\boldsymbol{H^{(0)}} = \boldsymbol{0}_{V \times h}$
5: **for** $i = 1, 2, ..., T$ **do**
6:     obtain dynamic variable subgraph $\boldsymbol{G^{(i)}} \in \mathbb{R}^{V \times V}$ according to $m^{(i)}$ and Eq.(8)
7:     encode observation values $x^{(i)}$ and timestamp $t^{(i)}$ into $\boldsymbol{X^{(i)}} \in \mathbb{R}^{V \times k}$ by multiple fully-connected mappings
8:     update $\boldsymbol{H^{(i)}} = \text{GCRNN}(\boldsymbol{\Theta}, \boldsymbol{G^{(i)}}, \boldsymbol{X^{(i)}}, \boldsymbol{H^{(i-1)}})$ using Eq.(10), (11), (12) and (11)
9: **end for**
10: calculate the sum of $h$ channels for $\boldsymbol{H^{(i)}}$ to get $C \in \mathbb{R}^V$
11: calculate $\hat{y} = Softmax(W^y C + b^y)$
12: **return** $\hat{y}$

---

# B  Datasets

We use four irregularly sampled medical time series datasets to evaluate the classification performance of our model and baseline models. The dataset statistics are summarized in Table 4.

Table 4: Dataset statistics.

| Datasets | P19 | P12 | MIMIC-III | Physionet |
|---|---|---|---|---|
| # Samples | 38803 | 11988 | 21107 | 3997 |
| # Variables | 34 | 36 | 16 | 36 |
| # Max Observation Length | 60 | 215 | 292 | 215 |
| Missing Ratio | 94.9% | 88.4% | 65.5% | 84.9% |

**P19**  The PhysioNet Sepsis Early Prediction Challenge 2019 [43] dataset consists of medical records from 38,803 patients. Each patient's record includes 34 variables of up to 60 hours and a static vector indicating attributes such as age, gender, the time interval between hospital admission and ICU admission, type of ICU, and length of stay in the ICU measured in days. Additionally, each patient is assigned a binary label indicating whether sepsis occurs within the subsequent 6 hours. We follow the procedures of [9] to ensure certain samples with excessively short or long time series are excluded. It is available at https://physionet.org/content/challenge-2019/1.0.0/.

**P12**  The P12 [46] dataset comprises data from 11,988 patients after 12 inappropriate samples identified by [5] were removed from the dataset. Each patient's record in the P12 dataset includes multivariate time series data collected during their initial 48-hour stay in the ICU. The time series data consists of measurements from 36 sensors (excluding weight). Additionally, each sample is associated with a static vector containing 9 elements, including age, gender, and other relevant attributes. Furthermore, each patient in the P12 dataset is assigned a binary label indicating the length of their stay in the ICU. A negative label signifies a hospitalization period of three days or shorter, while a positive label indicates a hospitalization period exceeding three days. It is available at https://physionet.org/content/challenge-2012/1.0.0/.

**MIMIC-III**  The MIMIC-III [45] dataset is a widely used database that comprises de-identified Electronic Health Records of patients who were admitted to the ICU at Beth Israel Deaconess

Medical Center from 2001 to 2012. It Originally encompassed around 57,000 records of ICU patients, containing diverse variables such as medications, in-hospital mortality, and vital signs. [48] established a variety of benchmark tasks using a subset of this database. In this paper, we focus on the binary in-hospital mortality prediction task to assess classification performance. Following preprocessing, our dataset consists of 16 features from the preceding 48 hours and 21,107 samples. It is available at https://physionet.org/content/mimiciii/1.4/.

**Physionet**   Physionet[44] contains the data from the first 48 hours of ICU patients, which is a reduced version of P12 considered by prior work. Therefore, we follow the same preprocessing methods used for the P12 dataset. The processed data set includes 3997 labeled instances. We focus on predicting in-hospital mortality. It is available at https://physionet.org/content/challenge-2012/.

## C   Baselines

The implementation of baselines follows the corresponding papers or default implementation of their code repositories. Considering that different baselines have varying convergence speeds, we search for the number of epochs in the range of $\{10, 20, 50\}$ to ensure all baselines can reach convergence. We search for the learning in the range of $\{0.001, 0.005\}$ due to the differences in model complexity and dataset size. Since all four datasets are highly imbalanced, we upsample the minority class in each batch to make the batch balance. Here are the detailed hyperparameter settings of these baselines:

**ODE-RNN**   [4]: ODE-RNN uses neural ODEs to model hidden state dynamics and an RNN to update the hidden state in the presence of a new observation. The latent dimension is set as $40$, and the ODE function has 3 layers with $50$ units. The source code can be found at https://github.com/YuliaRubanova/latent_ode

**GRU-D**   [17]: GRU-D is based on Gated Recurrent Unit (GRU), a state-of-the-art recurrent neural network. It takes two representations of missing patterns, i.e., masking and time interval, and effectively incorporates them into a deep model architecture. The number of hidden states of GRU-D is set as $49$. We use the open source code from https://github.com/Han-JD/GRU-D

**SEFT**   [5]: A set function approach where all the observations are modeled individually before pooling them together using an attention-based approach. We use a constant architecture for the attention network $f'$ with 2 layers, 4 heads and dimensionality of the dot product space $d$ of 128. In addition, the attention network $f'$ was always set to use mean aggregation. We use the open source code from https://github.com/BorgwardtLab/SeFT.

**mTAND**   [6]: A deep learning framework for ISMTS data that learns an embedding of continuous time values and uses an attention mechanism to produce a fixed-length representation. We use the encoder of the overall framework for classification tasks. We set the latent dimension and the hidden size of GRU to 32. The number of reference points and the dimension of time embedding is 128. The source code can be found at https://github.com/reml-lab/mTAN.

**IP-Net**   [14]: A model architecture for ISMTS data based on several semi-parametric interpolation layers organized into an interpolation network followed by a prediction network GRU. The number of reference points is set as $192$. The hidden size of GRU is $100$. We take the source code at https://github.com/mlds-lab/interp-net.

**DGM$^2$-O**   [13]: A generative model, which tracks the transition of latent clusters instead of isolated feature representations, achieves robust sparse time series modeling. We use the $\text{DGM}^2$-O and set both the hidden dimension and the cluster_num as 10. We use the source code at https://github.com/thuwuyinjun/DGM2.

**MTGNN**   [47]: A general graph neural network framework designed for MTS data. We use 5 graph convolution and 5 temporal convolution modules with the dilation exponential factor 2. The graph convolution and temporal convolution modules have 16 output channels. The skip connection layers all have 32 output channels. The first layer of the output module has 64 output channels, and the second layer has 1 output channel. We use the open source code from: https://github.com/nnzhan/MTGNN.

**Raindrop** [9]: A graph neural network that embeds ISMTS while learning the dynamics of sensors purely from observation data. The dimension of observation embedding is 4. The dimensions of time representation $p^t$ and $r_v$ are 16. We set the number of Raindrop layers $L$ as 2. The $d_k$ is set to 20. The $d_a$ is set equal to the number of sensors. The source code can be found at https://github.com/mims-harvard/Raindrop

**StraTS** [18] is a self-supervised transformer for sparse IMTS. We use the implementation at https://github.com/sindhura97/STraTS and the following setting in our experiment: hidden_dim = 64, num_layers = 2, num_heads = 16, dropout = 0.2.

**DuETT** [19] is a dual event time transformer for Electronic Health Records (EHRs). We use the implementation at https://github.com/layer6ai-labs/DuETT and the default settings of the model declaration in this repository.

**ViTST** [20] transforms IMTS into line graph images and adapts powerful vision transformers to perform time series classification in the same way as image classification. We use the implementation at https://github.com/Leezekun/ViTST.

**Warpformer** [22]: A transformer-based network that captures features at different scales in IMTS using warping modules and dual attention mechanisms. We use three scales with normalized length $\widetilde{L}^{(0)} = 0$, $\widetilde{L}^{(1)} = 0.2$ and $\widetilde{L}^{(2)} = 1$. The dimension of representations $D$ is set as 32. The attention heads and the layers of the warpformer are set as 1 and 2, respectively. We use the implementation at https://github.com/imJiawen/Warpformer.

## D   Performance Metrics

**AUROC**   AUROC is commonly employed in binary classification tasks, where one class is designated as positive and the other as negative. It represents the area under the Receiver Operating Characteristic (ROC) curve, constructed by plotting the True Positive Rate (TPR) against the False Positive Rate (FPR). AUROC ranges from 0 to 1, with a higher value indicating better model performance in accurately discriminating between positive and negative instances. An AUROC equal to 0.5 indicates a model's performance equivalent to random guessing, while an AUROC greater than 0.5 signifies superiority over random guessing.

**AUPRC**   The Area Under the Precision-Recall Curve is widely used as a performance metric for imbalanced binary classification tasks. It provides a comprehensive assessment of a model's precision-recall trade-off. The Precision-Recall curve is constructed by plotting recall on the x-axis and precision on the y-axis. AUPRC ranges from 0 to 1, and a higher value indicates better model performance in achieving high precision and recall simultaneously. It has been suggested as a good criterion for unevenly distributed classification problems [54].

## E   More Implementation Details

**Generation of Variables' Textual Descriptions**   We have chosen three textual sources: the full name of the variable, Wikipedia, and ChatGPT, for variable descriptions. The full name of the variable and Wikipedia sources can be directly extracted from dataset descriptions and web pages without requiring special processing, so we don't delve into them further here. Regarding the ChatGPT source, the medical properties description for each variable is generated by providing consistent question templates to ChatGPT, aiming to maintain a relatively uniform format across variables. We initiate queries to ChatGPT using the following template: *What medical prior knowledge do you know about the medical variable* ① *in* ②*?* to obtain a textual description for each variable in a specific dataset. The position marked as '①' corresponds to the variable name to be queried, and the position marked as '②' corresponds to the dataset-specific task description. For instance, the task for the MIMIC-III dataset is *ICU patients' in-hospital mortality prediction*, so the query for obtaining textual information associated with the variable *Heart Rate* could be: *What medical prior knowledge do you know about the medical variable Heart Rate in diagnosing whether ICU patients will die during hospitalization?* The following is a part of ChatGPT's answer for this example:

*Heart rate is a fundamental physiological variable that plays a significant role in diagnosing and assessing the condition of ICU patients and their potential risk of mortality during hospitalization. Here's what you need to know about the medical variable "heart rate" in this context: Normal Heart Rate: A normal resting heart rate for adults typically ranges between 60 to 100 beats per minute (bpm). Deviations from this range can indicate potential health issues. Tachycardia: An elevated heart rate (tachycardia), often above 100 bpm, can be a sign of various medical conditions ... .*

The complete text description can be obtained through the code link.

**Hyperparameters** We search all hyperparameters in the grid to find the best hyperparameters for our proposed model KEDGN. Specifically, our model has a total of 5 hyperparameters: dimension of query vectors $q$, dimension of variables' node embeddings $n$, proportion of density score $\alpha$, dimension of variables' hidden state $h$, and dimension of structured encoding representations $k$. For all datasets, $h$ and $k$ are set to be equal, and we search them over the range $\{8, 12, 16\}$. Additionally, we search the dimension of query vectors $q$ in $\{5, 7, 9\}$, the dimension of variables' node embedding $n$ in $\{7, 9, 11\}$ and the proportion of density score $\alpha$ in $\{1.0, 2.0, 3.0\}$. The best hyperparameters for each dataset are reported in the code.

# F    Additional Experiment

## F.1    Classification Results Using Different PLMs

In KEDGN, we utilize a PLM to extract sentence embeddings of variables' textual medical information as variable semantic representations. The choice of PLM is diverse, and in our default implementation, we use BERT. Additionally, we have experimented with other PLMs, including T5 [55], Bart [56], GPT2 [57], LED [58], and Pegasus [59]. The detailed information of these PLMs is listed in Table 5. We extract the hidden state at

Table 5: Details of Pre-trained Language Models.

| Model | Size | HuggingFace Key | Architecture |
| --- | --- | --- | --- |
| T5 | 223M | t5-base | Encoder-Decoder |
| Bart | 139M | facebook/bart-base | Encoder-Decoder |
| GPT2 | 137M | gpt2 | Decoder-Only |
| LED | 139M | allenai/led-base-16384 | Encoder-Decoder |
| Pegasus | 568M | google/pegasus-xsum | Encoder-Decoder |
| BERT | 110M | bert-base-uncased | Encoder-Only |

the [CLS] position for BERT. For other models, we uniformly take the last hidden state (average pooling) of the models as the representations of the variables. The classification results of using different PLMs on four datasets are listed in Table 6.

The results show that BERT (Encoder-Only model) achieved high performance on most datasets. Several Encoder-Decoder-based models also demonstrated competitive results, with Bart outperforming BERT on the P12 dataset, LED slightly surpassing BERT on the Physionet dataset and T5 showing slightly lower performance than BERT on the MIMIC-III dataset. However, the Decoder-Only GPT exhibited notably lower results on the P19 dataset and MIMIC-III compared to other models. This situation may be attributed to the fact that, in our task, PLMs are used to extract sentence-level semantic representations, which are utilized to differentiate variables and measure correlations among them, resembling a text classification process among variables. On the one hand, Encoder-Only models may excel at understanding tasks such as text classification, with BERT specifically adding a special [CLS] token dedicated to extracting sentence-level overall semantic information. On the other hand, models incorporating a decoder involve tasks related to predicting the next word and may be more focused on text generation tasks. They may exhibit a slight deficiency in extracting distinctive sentence-level representations, especially for Decoder-Only models. These models sometimes require carefully designed prompts to guide them in generating high-quality outputs.

In summary, variable semantic representations extracted by different PLMs can impact downstream task performance. Choosing a PLM that is more suitable for the task and dataset can further enhance the performance of KEDGN. In the context of the application in this paper, a simple pure encoder model may be more suitable for achieving optimal results.

## F.2    Computational Cost Analysis

Table 6: Classification results using different PLMs. The best results are highlighted in **bold**, and the second-best results are in underlined. The results in the table are presented in the form of (Mean ± Std %).

| | P19 | | Physionet | | MIMIC-III | | P12 | |
|---|---|---|---|---|---|---|---|---|
| | AUROC | AUPRC | AUROC | AUPRC | AUROC | AUPRC | AUROC | AUPRC |
| T5 | 91.4 ± 0.8 | 59.7 ± 1.2 | 87.3 ± 0.5 | 56.9 ± 2.5 | 84.6 ± 0.3 | 48.0 ± 0.8 | 87.5 ± 0.4 | 53.6 ± 2.3 |
| Bart | 91.6 ± 0.6 | 59.8 ± 1.6 | 85.7 ± 1.3 | 53.3 ± 2.5 | 82.5 ± 2.9 | 44.4 ± 4.6 | 87.7 ± 0.6 | **55.0 ± 2.0** |
| GPT2 | 91.4 ± 1.6 | 54.8 ± 7.5 | 87.4 ± 0.7 | 56.1 ± 1.6 | 82.6 ± 2.0 | 43.1 ± 4.0 | 87.4 ± 0.2 | 53.2 ± 1.7 |
| LED | 91.7 ± 0.5 | 60.3 ± 0.8 | 87.5 ± 0.6 | **57.3 ± 2.5** | 84.5 ± 0.4 | 47.9 ± 0.8 | 86.8 ± 0.5 | 51.1 ± 1.6 |
| Pegasus | 91.9 ± 0.5 | 61.1 ± 1.2 | 87.1 ± 1.2 | 52.8 ± 2.5 | 84.4 ± 0.8 | 48.0 ± 1.9 | 87.3 ± 0.5 | 53.7 ± 1.1 |
| BERT(Default) | **92.2 ± 0.5** | **62.0 ± 1.3** | **87.9 ± 0.2** | 57.1 ± 1.8 | **85.1 ± 0.3** | **48.3 ± 1.6** | **87.8 ± 0.5** | 54.5 ± 1.5 |

We conduct a analysis of the time and space overhead on the Physionet dataset, with a batch size of 128, and utilizing Nvidia 1080Ti GPU infrastructure. The results are shown in Table 7. Our method achieves a balanced time and space overhead. The introduction of textual information involves generating semantic embeddings using PLMs, and the adjustment of the variable graph based on observed local density, both of which can be predetermined and integrated into the preprocessing step without increasing model training overhead. Attention-based methods (SeFT, mTAND) achieve parallel computing in the time dimension, resulting in low time overhead, but this sacrifices fine-grained feature extraction at the variable level. Furthermore, compared to the same RNN-based method GRU-D, our method only deals with actual observation points at each timestamp, thus significantly reducing time overhead. While our method's runtime is 1.3 times that of the latest SOTA model, Warpformer, our space overhead is only 16% of its size. In situations where computational resources permit, our method can further reduce runtime by employing the space-for-time trade-off strategy (such as increasing batch size).

Table 7: Comparison of computational costs on Physionet dataset.

| Model | time(min/epoch) | space(MiB) |
|---|---|---|
| ODE-RNN | 5.06 | 2582 |
| GRU-D | 1.32 | 796 |
| SeFT | 0.07 | 684 |
| mTAND | 0.05 | 4658 |
| DGM$^2$-O | 0.06 | 688 |
| Raindrop | 0.17 | 4864 |
| Warpformer | 0.33 | 11084 |
| **KEDGN** | 0.44 | 1798 |

## F.3 More Results for Leave-variables-out

Table 8: Classification performance on samples with a fixed set of left-out variables. The best results are highlighted in **bold** and the second best results are in underlined.

| Dataset | Methods | Discard ratio | | | | | | | | | |
| | | 10% | | 20% | | 30% | | 40% | | 50% | |
| | | AUROC | AUPRC | AUROC | AUPRC | AUROC | AUPRC | AUROC | AUPRC | AUROC | AUPRC |
|---|---|---|---|---|---|---|---|---|---|---|---|
| P12 | GRU-D | 68.6 ± 2.3 | 35.8 ± 2.2 | 68.2 ± 2.1 | 34.5 ± 2.9 | 66.8 ± 3.3 | 32.7 ± 4.6 | 65.8 ± 4.0 | 31.3 ± 5.2 | 65.1 ± 4.1 | 30.4 ± 5.5 |
| | mTAND | 74.9 ± 0.6 | 37.7 ± 0.6 | 74.0 ± 1.3 | 36.5 ± 1.5 | 71.4 ± 3.8 | 34.1 ± 3.7 | 70.6 ± 3.6 | 33.2 ± 3.7 | 70.1 ± 3.5 | 32.5 ± 3.6 |
| | DGM$^2$-O | 76.3 ± 1.1 | 39.3 ± 1.5 | 76.1 ± 1.1 | 38.2 ± 1.7 | 74.8 ± 2.2 | 36.8 ± 2.6 | 72.0 ± 5.3 | 34.3 ± 5.0 | 70.4 ± 5.9 | 32.7 ± 5.7 |
| | MTGNN | 71.2 ± 2.1 | 30.5 ± 1.5 | 70.3 ± 3.3 | 29.7 ± 2.8 | 68.9 ± 4.2 | 28.5 ± 3.3 | 68.1 ± 4.7 | 27.7 ± 3.6 | 67.6 ± 5.2 | 27.2 ± 3.8 |
| | Raindrop | 73.2 ± 1.6 | 32.4 ± 0.9 | 73.0 ± 1.6 | 31.7 ± 1.1 | 72.2 ± 2.6 | 31.1 ± 2.7 | 71.5 ± 3.5 | 30.6 ± 3.5 | 70.8 ± 4.2 | 29.7 ± 4.3 |
| | DuETT | 73.9 ± 1.7 | 35.8 ± 2.3 | 74.7 ± 1.8 | 35.3 ± 2.0 | 73.6 ± 2.2 | 34.1 ± 2.4 | 72.8 ± 2.6 | 33.3 ± 2.7 | 72.3 ± 2.7 | 32.6 ± 2.8 |
| | Warpformer | 75.9 ± 0.7 | 37.3 ± 2.2 | 75.6 ± 0.8 | 36.7 ± 2.3 | 73.8 ± 2.9 | 34.3 ± 4.1 | 72.8 ± 3.4 | 33.0 ± 4.6 | 72.1 ± 3.7 | 32.2 ± 4.7 |
| | **Ours** | **79.7 ± 0.4** | **43.6 ± 1.2** | **79.2 ± 0.8** | **42.5 ± 1.6** | **77.7 ± 2.2** | **40.0 ± 4.0** | **77.2 ± 2.2** | **39.6 ± 3.7** | **76.9 ± 2.2** | **39.2 ± 3.5** |
| P19 | GRU-D | 88.5 ± 2.3 | 54.6 ± 3.7 | 88.8 ± 2.1 | 54.2 ± 3.4 | 88.0 ± 2.5 | 50.4 ± 7.5 | 87.5 ± 2.8 | 49.6 ± 6.9 | 86.4 ± 3.5 | 47.2 ± 8.6 |
| | mTAND | 79.6 ± 1.8 | 28.6 ± 1.9 | 79.2 ± 1.9 | 28.1 ± 2.1 | 78.0 ± 2.4 | 26.9 ± 2.9 | 77.2 ± 2.7 | 26.3 ± 2.9 | 76.2 ± 3.2 | 24.3 ± 4.8 |
| | DGM$^2$-O | 87.4 ± 0.6 | 53.4 ± 1.5 | 87.3 ± 0.8 | 53.2 ± 1.7 | 86.6 ± 1.6 | 49.9 ± 5.1 | 85.8 ± 1.9 | 47.7 ± 5.9 | 85.2 ± 2.2 | 45.7 ± 6.7 |
| | MTGNN | 84.5 ± 1.4 | 48.9 ± 2.3 | 84.8 ± 1.7 | 49.8 ± 3.1 | 84.0 ± 1.9 | 47.2 ± 4.8 | 83.3 ± 2.2 | 45.5 ± 5.5 | 82.5 ± 2.9 | 42.7 ± 9.2 |
| | Raindrop | 88.2 ± 1.5 | 59.7 ± 1.5 | 88.1 ± 1.3 | **59.8 ± 1.4** | 87.8 ± 1.2 | 59.1 ± 1.7 | 87.6 ± 1.1 | 58.5 ± 1.9 | 87.1 ± 1.5 | 57.7 ± 2.3 |
| | DuETT | 85.2 ± 1.0 | 53.7 ± 1.0 | 84.8 ± 1.1 | 53.9 ± 0.8 | 84.7 ± 1.0 | 53.3 ± 1.6 | 84.3 ± 1.4 | 52.7 ± 2.1 | 84.4 ± 1.3 | 52.5 ± 2.0 |
| | Warpformer | 91.3 ± 0.8 | 55.2 ± 5.6 | **91.3 ± 0.8** | 55.1 ± 5.6 | **91.4 ± 0.8** | 56.0 ± 4.8 | **91.5 ± 0.7** | 56.4 ± 4.3 | **91.2 ± 0.8** | 56.2 ± 3.9 |
| | **Ours** | **91.3 ± 0.3** | **59.9 ± 0.7** | 91.2 ± 0.5 | 59.6 ± 0.9 | 90.9 ± 0.9 | **59.1 ± 1.1** | 90.7 ± 1.0 | **58.9 ± 1.2** | 90.1 ± 1.6 | **58.1 ± 2.0** |
| Physionet | GRU-D | 70.0 ± 3.0 | 32.1 ± 4.1 | 69.5 ± 2.6 | 31.1 ± 3.6 | 69.2 ± 3.0 | 31.0 ± 4.4 | 68.3 ± 3.6 | 30.1 ± 5.3 | 68.1 ± 3.7 | 29.8 ± 5.3 |
| | mTAND | 80.5 ± 2.1 | 42.8 ± 4.0 | 78.2 ± 3.4 | 40.5 ± 4.7 | 76.3 ± 4.0 | 37.7 ± 5.7 | 75.6 ± 3.9 | 36.6 ± 5.6 | 75.1 ± 3.9 | 36.1 ± 5.1 |
| | DGM$^2$-O | 80.2 ± 0.9 | 38.6 ± 2.8 | 80.4 ± 0.9 | 38.3 ± 2.8 | 79.3 ± 1.9 | 37.1 ± 3.4 | 77.5 ± 3.7 | 35.4 ± 4.4 | 75.6 ± 5.0 | 34.0 ± 4.8 |
| | MTGNN | 68.9 ± 4.1 | 25.8 ± 4.8 | 69.3 ± 4.3 | 26.6 ± 4.5 | 69.0 ± 4.8 | 26.3 ± 5.2 | 68.3 ± 5.2 | 25.4 ± 4.8 | 67.2 ± 5.4 | 24.4 ± 4.8 |
| | Raindrop | 76.5 ± 1.2 | 33.4 ± 2.2 | 76.5 ± 1.3 | 32.3 ± 2.3 | 75.6 ± 2.0 | 30.8 ± 3.2 | 74.7 ± 2.6 | 29.7 ± 3.5 | 73.6 ± 3.2 | 28.8 ± 3.9 |
| | DuETT | 78.2 ± 2.8 | 39.9 ± 3.5 | 78.3 ± 3.0 | 39.9 ± 3.7 | 76.7 ± 3.7 | 37.9 ± 4.5 | 75.9 ± 3.8 | 37.0 ± 4.6 | 74.9 ± 4.3 | 35.9 ± 5.0 |
| | Warpformer | 78.2 ± 1.0 | 33.3 ± 2.1 | 77.7 ± 1.6 | 33.6 ± 1.8 | 75.8 ± 3.4 | 31.8 ± 3.0 | 73.8 ± 4.6 | 30.2 ± 4.1 | 72.7 ± 4.9 | 29.2 ± 4.2 |
| | **Ours** | **83.8 ± 1.0** | **49.4 ± 2.5** | **82.9 ± 2.5** | **48.0 ± 5.3** | **81.7 ± 2.8** | **46.4 ± 5.5** | **81.4 ± 2.5** | **45.8 ± 5.2** | **81.1 ± 2.4** | **45.2 ± 5.3** |

## F.4 More Results for Ablation Study

Table 9: The ablation study of our proposed method KEDGN. The results in the table are presented in the form of (Mean ± Std %).

| | Physionet | | MIMIC-III | | P12 | |
|---|---|---|---|---|---|---|
| | AUROC | AUPRC | AUROC | AUPRC | AUROC | AUPRC |
| w/o VSW | 86.6 ± 1.2 | 51.1 ± 3.2 | 84.6 ± 0.5 | 47.6 ± 0.4 | 86.1 ± 0.9 | 50.8 ± 1.2 |
| w/o Text | 87.7 ± 0.4 | 54.3 ± 2.0 | 83.4 ± 0.7 | 44.5 ± 1.5 | 87.7 ± 0.5 | **54.9 ± 0.9** |
| w/o Graph | 86.8 ± 0.7 | 55.2 ± 1.6 | 84.5 ± 0.4 | 48.0 ± 0.5 | 87.8 ± 0.6 | 53.2 ± 1.3 |
| w/o KEE | 86.6 ± 1.2 | 53.3 ± 1.8 | 83.9 ± 0.7 | 46.1 ± 1.9 | 87.5 ± 0.4 | 53.3 ± 0.5 |
| w/o DAG | 87.2 ± 0.9 | 55.8 ± 2.4 | 84.7 ± 0.8 | 47.7 ± 2.3 | 87.4 ± 0.1 | 52.1 ± 1.1 |
| w/o TE | 86.7 ± 1.9 | 55.2 ± 3.6 | 84.2 ± 0.4 | 47.3 ± 1.6 | 87.3 ± 0.4 | 53.9 ± 1.8 |
| **Full** | **87.9 ± 0.2** | **57.1 ± 1.8** | **85.1 ± 0.3** | **48.3 ± 1.6** | **87.8 ± 0.5** | 54.5 ± 1.5 |

We find that the model performance declines after introducing text information on P12 dataset. As indicated in Table 6, when we replace PLM from BERT to Bart, there is a slight improvement of 0.5% in AUPRC on the P12 dataset, slightly outperforming the model without text. Therefore, the reason for the decline in model performance is likely attributed to the choice of default PLM (BERT), which may not be optimal for extracting textual embeddings on this dataset, rather than the text itself causing the decline. In other words, introducing text offers the potential for enhancing model performance, with the extent of improvement depending on the degree of effective utilization of text.

## F.5 Effects of different activation functions in Eq.(7)

Table 10: Comparison of the results of different activation functions in Eq.(7).

| | P19 | | Physionet | | MIMIC-III | | P12 | |
|---|---|---|---|---|---|---|---|---|
| | AUROC | AUPRC | AUROC | AUPRC | AUROC | AUPRC | AUROC | AUPRC |
| w/o $\sigma$ | 90.3 ± 0.9 | 58.1 ± 1.8 | 86.2 ± 0.8 | 52.8 ± 4.5 | 84.3 ± 0.6 | 47.3 ± 1.5 | 86.8 ± 0.5 | 51.0 ± 1.2 |
| $\sigma$ Sigmoid | 91.6 ± 0.8 | 59.6 ± 2.6 | 87.2 ± 0.7 | 56.0 ± 1.6 | 84.4 ± 0.7 | 47.5 ± 1.8 | 86.9 ± 0.8 | 52.9 ± 2.2 |
| $\sigma$ Tanh (Default) | **92.2 ± 0.5** | **62.0 ± 1.3** | **87.9 ± 0.2** | **57.1 ± 1.8** | **85.1 ± 0.3** | **48.3 ± 1.6** | **87.8 ± 0.5** | **54.5 ± 1.5** |

On one hand, the activation function in Eq.(7) reflects the dynamics of variables, such as time decay or exponential increase. On the other hand, this activation function serves a normalization purpose because the edge weights in the knowledge-empowered complete graph $A$ are normalized values. If directly using the absolute value of the density to adjust the edge weights, the values might become excessively large or small, which would severely disrupt the basic graph structure learned from textual knowledge. Typical activation functions with normalization capabilities include Sigmoid and Tanh. As shown in Table 10, Tanh is chosen since it performs better.

## F.6 Parameter Complexity Analysis

Table 11: Comparison of the number of model parameters for three models.

| Datasets | P19 | Physionet | MIMIC-III | P12 |
|---|---|---|---|---|
| Raindrop | 1947668 | 19789024 | 35647028 | 19789024 |
| Warpformer | 43780 | 43974 | 42034 | 43974 |
| Ours | 80262 | 46545 | 51129 | 63087 |

As shown in Table 11, the parameter count of our model is not particularly high. It is on the same order of magnitude as Warpformer and significantly lower than Raindrop by three orders of magnitude. Although we calculate an independent parameter space for each variable, the total $W_t \in \mathbb{R}^{V \times I \times O}$ does not equate to the parameter count. $W_t$ is derived from the multiplication of two matrices: the variable embedding matrix $Q \in \mathbb{R}^{V \times q}$ and the weight matrix $W \in \mathbb{R}^{q \times I \times O}$. The first matrix is computed from textual embeddings, and only the second matrix belongs to the model parameters. The sizes $q$, $I$, and $O$ are hyperparameters, independent of the number of variables, and typically set to be less than 16. This ensures that the parameter complexity of our model remains within an acceptable range.

### F.7 More variables groups on the P12 dataset

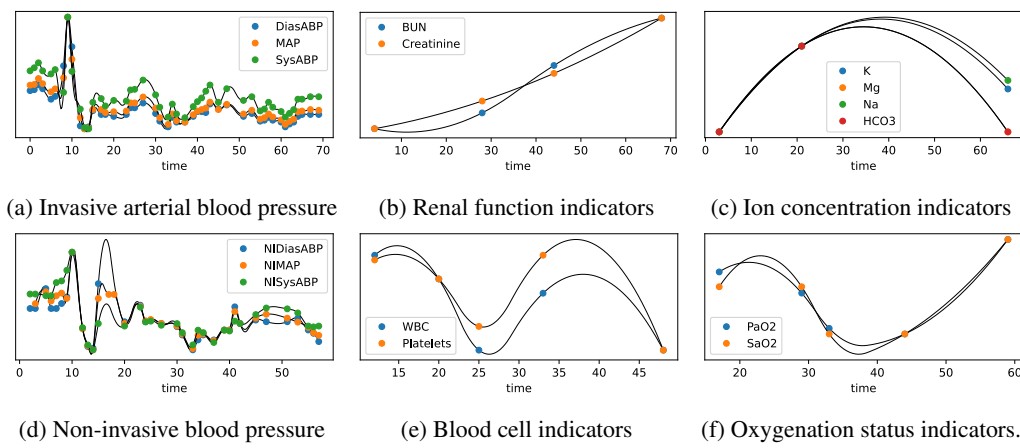

(a) Invasive arterial blood pressure

(b) Renal function indicators

(c) Ion concentration indicators

(d) Non-invasive blood pressure

(e) Blood cell indicators

(f) Oxygenation status indicators.

Figure 7: Variable groups (Partial) divided by temporal patterns on the P12 dataset.

### F.8 More Visualizations of Variables Textual Representations

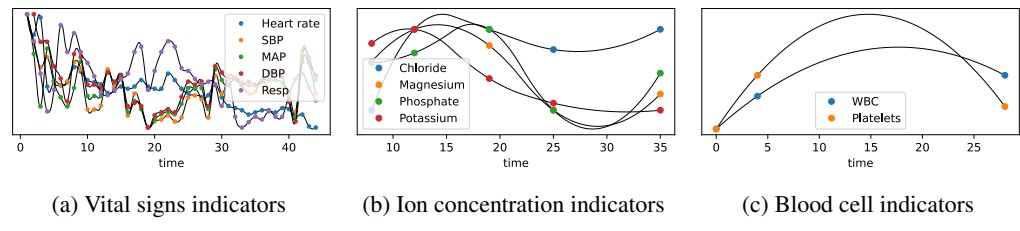

(a) Vital signs indicators

(b) Ion concentration indicators

(c) Blood cell indicators

Figure 8: Variable groups (Partial) divided by temporal patterns on the P19 dataset.

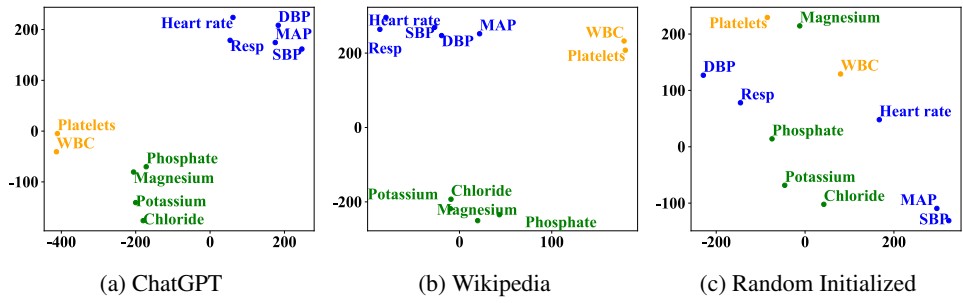

(a) ChatGPT

(b) Wikipedia

(c) Random Initialized

Figure 9: T-SNE visualization of partial variable semantic representations on the P19 dataset.

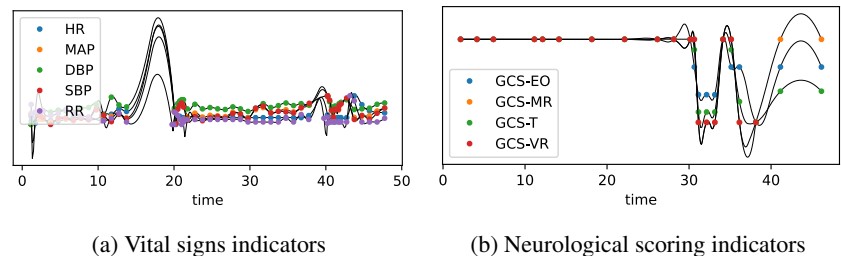

(a) Vital signs indicators

(b) Neurological scoring indicators

Figure 10: Variable groups (Partial) divided by temporal patterns on the MIMIC-III dataset.

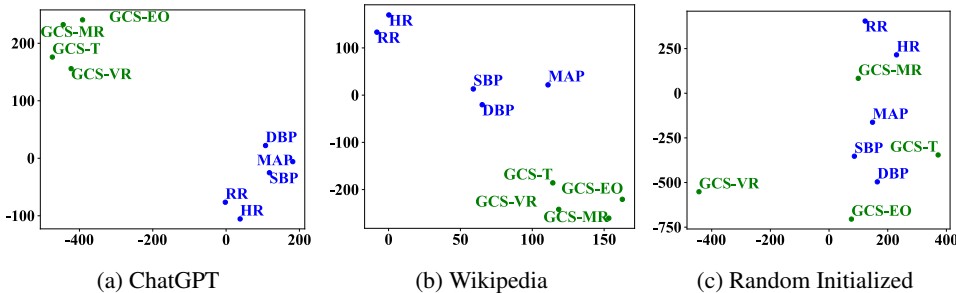

(a) ChatGPT  (b) Wikipedia  (c) Random Initialized

Figure 11: T-SNE visualization of partial variable semantic representations on the MIMIC-III dataset.

## F.9 Visualization of Variable Semantic Representations Using Different PLMs

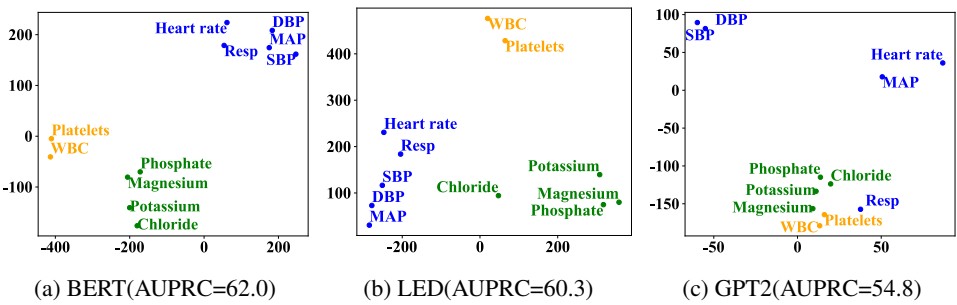

(a) BERT(AUPRC=62.0)  (b) LED(AUPRC=60.3)  (c) GPT2(AUPRC=54.8)

Figure 12: T-SNE visualization of variable semantic representations generated by different PLMs on the P19 dataset.

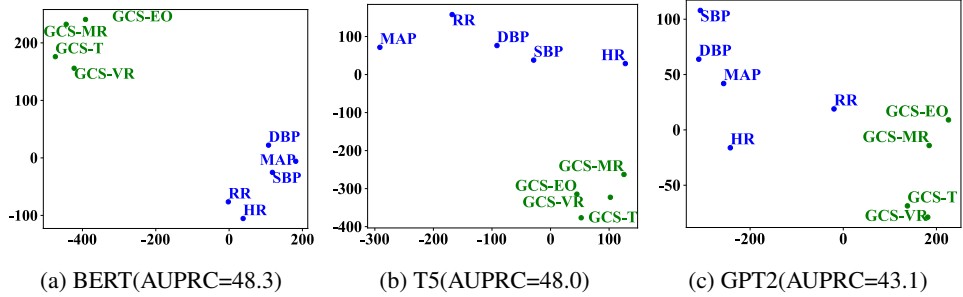

(a) BERT(AUPRC=48.3)  (b) T5(AUPRC=48.0)  (c) GPT2(AUPRC=43.1)

Figure 13: T-SNE visualization of variable semantic representations generated by different PLMs on the MIMIC-III dataset.

We conduct T-SNE visualization analysis on variable representations obtained from different PLMs. Figures 12 and 13 correspond to the results of the P19 and MIMIC-III datasets, respectively. We observe that when the clusters of variable representations extracted by PLMs are consistent with the grouping of variable time series patterns and exhibit distinctiveness, the corresponding downstream classification task performance tends to be better. For example, on the P19 dataset, BERT and LED achieve higher classification performance, with their corresponding variable representations having good distinctiveness: the three groups of variables represented by blue, yellow, and green colors show high cohesion and low coupling, while the T5 model, which exhibits suboptimal classification performance, has a distribution where Resp, WBC, and Platelets are notably confused with the green variables. Additionally, in the MIMIC-III dataset, the variable representations obtained from BERT and T5, which achieve higher performance, with blue and green clusters having longer distances, indicating better distinctiveness. On the other hand, variable representations obtained from GPT2 show poor distinctiveness, with the variable RR being close to GCS-EO and GCS-MR.

