# OpenReview forum: "Knowledge-Empowered Dynamic Graph Network for Irregularly Sampled Medical Time Series"
_NeurIPS.cc/2024/Conference — NeurIPS 2024 poster_

### Official Review · Reviewer_n8KK · 2024-07-08

**Soundness:** 3
**Presentation:** 3
**Contribution:** 3
**Rating:** 6
**Confidence:** 4

**Summary:**

The paper aims to handle difference and correlation between multiple variables in the irregularly sampled medical time series data. By proposing a model named KEDGN with textual medical knowledge and dynamic variable graphs, it is able to outperforms baselines on a variety of datasets. Comprehensive experiments have also been conducted to demonstrate the learned correlations between variables, which aligns with the motivation of this paper.

**Strengths:**

1.	The motivation of the paper is solid. The authors provide enough evidence and arguments to demonstrate the dynamic correlation patterns between time series data.
2.	The paper is generally well written with clear figures and equations.
3.	The proposed model demonstrates strong performance in various experiment settings. Comprehensive analysis is also being conducted to prove the effectiveness of each module design.
4.	It is good to see some visualizations about the variable correlations learned by the model, which aligns with the motivation of this paper.

**Weaknesses:**

1.	The idea of learning dynamic graphs that model variable relationships is quite common in literatures. It would be better for the authors to cite related works, compare with existing works and make arguments about novelty and effectiveness of the proposed modules. Here are some references:
    -	Jiang, Renhe, Zhaonan Wang, Jiawei Yong, Puneet Jeph, Quanjun Chen, Yasumasa Kobayashi, Xuan Song, Shintaro Fukushima, and Toyotaro Suzumura. "Spatio-temporal meta-graph learning for traffic forecasting." In Proceedings of the AAAI conference on artificial intelligence, vol. 37, no. 7, pp. 8078-8086. 2023.
    -	Huang, Qihe, Lei Shen, Ruixin Zhang, Shouhong Ding, Binwu Wang, Zhengyang Zhou, and Yang Wang. "Crossgnn: Confronting noisy multivariate time series via cross interaction refinement." Advances in Neural Information Processing Systems 36 (2023): 46885-46902.
    -	Wang, Dingsu, Yuchen Yan, Ruizhong Qiu, Yada Zhu, Kaiyu Guan, Andrew Margenot, and Hanghang Tong. "Networked time series imputation via position-aware graph enhanced variational autoencoders." In Proceedings of the 29th ACM SIGKDD Conference on Knowledge Discovery and Data Mining, pp. 2256-2268. 2023.
    -	Wang, Binwu, Pengkun Wang, Yudong Zhang, Xu Wang, Zhengyang Zhou, Lei Bai, and Yang Wang. "Towards Dynamic Spatial-Temporal Graph Learning: A Decoupled Perspective." In Proceedings of the AAAI Conference on Artificial Intelligence, vol. 38, no. 8, pp. 9089-9097. 2024.
2.	There exist some missing references and baselines for irregularly sampled time series classification problems. It would be better to cite related works and compare with them in the experiment sections. Here are some examples:
    -	Li, Zekun, Shiyang Li, and Xifeng Yan. "Time Series as Images: Vision Transformer for Irregularly Sampled Time Series." Advances in Neural Information Processing Systems 36 (2024).
    -	Labach, Alex, Aslesha Pokhrel, Xiao Shi Huang, Saba Zuberi, Seung Eun Yi, Maksims Volkovs, Tomi Poutanen, and Rahul G. Krishnan. "Duett: Dual event time transformer for electronic health records." In Machine Learning for Healthcare Conference, pp. 403-422. PMLR, 2023.

**Questions:**

1.	It seems some of the baselines such as DGM^2-O can reach good performance with relatively less time and space requirements. Can there be any potential improvements to the efficiency of the proposed KEDGN model? It seems BERT is also optimized during training, it would be interesting to see some efficiency experiments and analysis regarding this aspect.

---

> ### Author Rebuttal · Authors · 2024-08-06
>
> # Responses to Reviewer n8KK (1/1)
>
> Dear reviewer n8KK,
>
> Thank you for your valuable feedback on the supplement of relevant literature of the article. Here is the response:
>
> **W1. The idea of learning dynamic graphs that model variable relationships is quite common in literatures. It would be better for the authors to cite related works, compare with existing works and make arguments about novelty and effectiveness of the proposed modules.**
>
> - "Spatio-temporal meta-graph learning for traffic forecasting." In Proceedings of the AAAI conference on artificial intelligence, vol. 37, no. 7, pp. 8078-8086. 2023.
> - "Crossgnn: Confronting noisy multivariate time series via cross interaction refinement." Advances in Neural Information Processing Systems 36 (2023): 46885-46902.
> - "Networked time series imputation via position-aware graph enhanced variational autoencoders." In Proceedings of the 29th ACM SIGKDD Conference on Knowledge Discovery and Data Mining, pp. 2256-2268. 2023.
> - "Towards Dynamic Spatial-Temporal Graph Learning: A Decoupled Perspective." In Proceedings of the AAAI Conference on Artificial Intelligence, vol. 38, no. 8, pp. 9089-9097. 2024.
>
> A1. Thank you for the reminder. In a later version, we will cite these four papers in the section on “Graph Neural Networks for Multivariate Time Series” in the related work. Although modeling dynamic graphs to capture relationships between variables is not new, many dynamic spatiotemporal graph methods are designed for traffic flow prediction tasks, which involve regularly sampled multivariate time series. Applying these methods directly to irregularly sampled medical time series classification tasks may lead to suboptimal performance. Our approach specifically addresses the challenges of ISMTS by using domain knowledge as guidance and adjusting edge weights in the graph based on observation masks and dynamic densities, which is a key distinction from these methods.
>
>
>
> **W2. There exist some missing references and baselines for irregularly sampled time series classification problems. It would be better to cite related works and compare with them in the experiment sections.**
>
> - Li, Zekun, Shiyang Li, and Xifeng Yan. "Time Series as Images: Vision Transformer for Irregularly Sampled Time Series." Advances in Neural Information Processing Systems 36 (2023).
> - Labach, Alex, Aslesha Pokhrel, Xiao Shi Huang, Saba Zuberi, Seung Eun Yi, Maksims Volkovs, Tomi Poutanen, and Rahul G. Krishnan. "Duett: Dual event time transformer for electronic health records." In Machine Learning for Healthcare Conference, pp. 403-422. PMLR, 2023.
>
> A2. Thank you for the reminder. In the revised version, we will cite these two papers in the section on “Irregularly Sampled Multivariate Time Series Modeling” in the related work. Additionally, we have conducted comparative experiments, and the results are as follows:
>
> |            | P19      |          | Physionet |          | MIMIC-III |          | P12      |          |
> | ---------- | -------- | -------- | --------- | -------- | --------- | -------- | -------- | -------- |
> |            | AUROC    | AUPRC    | AUROC     | AUPRC    | AUROC     | AUPRC    | AUROC    | AUPRC    |
> | ViTST      | 91.7±0.1 | 57.5±0.7 | 81.3±1.9  | 37.4±2.9 | 81.8±0.3  | 39.6±1.3 | 86.3±0.1 | 50.8±1.5 |
> | Duett      | 88.2±0.5 | 56.0±3.9 | 81.3±1.4  | 44.9±1.4 | 78.8±0.8  | 34.3±1.0 | 83.4±1.2 | 45.4±1.5 |
> | Ours(Best) | 92.3±1.0 | 62.5±0.7 | 88.2±1.1  | 57.5±2.5 | 85.1±0.3  | 48.4±1.5 | 87.8±0.5 | 54.5±1.5 |
>
>
>
>
>
> **Q1. It seems some of the baselines such as DGM^2-O can reach good performance with relatively less time and space requirements. Can there be any potential improvements to the efficiency of the proposed KEDGN model? It seems BERT is also optimized during training, it would be interesting to see some efficiency experiments and analysis regarding this aspect.**
>
> A3. Firstly, it should be clarified that BERT is frozen and does not participate in the optimization of training. We only take the sentence representation output by BERT as the semantic representation of variables, so the training cost is consistent with the variable embedding of random initialization. In addition, due to the sequential nature of RNN computation, there is an inherent bottleneck in the running time of our model. We have optimized the code implementation as much as possible to keep the overall running cost of the model within an acceptable range. We acknowledge this as a limitation of our model and will add it to the limitations section.

---

> > ### Comment · Reviewer_n8KK · 2024-08-08
> > **Response to rebuttal**
> >
> > Thank you for the detailed response. All my concerns have been addressed properly and I'd like to keep my score.

---

### Official Review · Reviewer_68iN · 2024-07-13

**Soundness:** 3
**Presentation:** 3
**Contribution:** 3
**Rating:** 6
**Confidence:** 5

**Summary:**

This work considers the irregular timestamp contained in current medical data. They employ a density-aware mechanism to the time-varying correlations among variables.

**Strengths:**

This work addresses an important topic in this field and is well-organized and written. The experiments are sufficient to prove their conclusions.

**Weaknesses:**

1. In formulation (6), the density of the observation point is calculated by the average time interval. What if the point is not observed in the next or previous time? e.g. t_v^(i+1) or t_v^(i) is None.
2. I am just a little bit confused by the node in the dynamic variable graph. As we know, the patient's EHR data will contain lots of clinical notes and some like treatment, medications, and so on. You have shown the description in your model's framework in Figure 3. So do you consider other variables in patient's EHR data? Besides, the node here denotes a text, not detailed medical terminology, right?
3. Could you please add a comparison graph between the original variable representation and the final representations?

**Questions:**

see weakness

**Limitations:**

Yes

---

> ### Author Rebuttal · Authors · 2024-08-06
>
> # Responses to Reviewer 68iN (1/1)
>
> Dear reviewer 68iN,
>
> Thank you for your valuable feedback on the formula details and completeness of the paper. Here is the response:
>
> **W1. $Z^{t}$ 's definition is not clear**
>
> A1. We apologize for the oversight. If there are no observations at the preceding/following time points, we take the time interval of the following/preceding observation as the density. If neither a preceding nor a following observation exists, it indicates that this observation is the only one for the variable, so we take half of the maximum observation time span across all variables as the density.
>
> The corrected definition is as follows:
> $$
> Z^{(t)}=Z_{i, v}=\begin{cases}((t_{i,v}-t_{i-1,v}) + (t_{i+1,v}- t_{i, v})) / 2, \quad  \text{if }t_{i+1,v} \text{ and } t_{i-1,v} \text{ are both not None.}\\\\t_{i,v}-t_{i-1,v}, \quad  \text{if }t_{i+1,v}  \text{ is None.} \\\\t_{i+1,v}- t_{i, v}, \quad  \text{if } t_{i-1,v} \text{ is None.}\\\\t_{max} / 2, \quad  \text{if }t_{i+1,v} \text{ and } t_{i-1,v} \text{ are both None.}\end{cases}
> $$
> Here, $Z^{(t)} = Z_{i, v} $ represents the average density at time $ t $ for the $ v $-th variable's $ i $-th observation.
>
>
>
> **W2. I am just a little bit confused by the node in the dynamic variable graph. As we know, the patient's EHR data will contain lots of clinical notes and some like treatment, medications, and so on. You have shown the description in your model's framework in Figure 3. So do you consider other variables in patient's EHR data? Besides, the node here denotes a text, not detailed medical terminology, right?**
>
> A2. Here, we use general medical semantic text to differentiate between variables and guide the modeling of correlations between these variables in irregular time series. If I understand correctly, the "other variables in patient's EHR data" you mentioned likely refer to patient-specific clinical records, treatment history, medication history, etc. These are personalized medical data and not general medical domain knowledge. They often include multiple modalities, such as text and images, which extend beyond the scope of time series analysis. Our datasets only contain time series data and do not include clinical notes. Thank you for the constructive suggestion; we may consider integrating multimodal medical data in future work.
>
>
>
> **W3. Could you please add a comparison graph between the original variable representation and the final representations?**
>
> A3. Certainly, but due to the limited information that can be uploaded during discussion period, we will include this in a later version.

---

> > ### Comment · Reviewer_68iN · 2024-08-09
> >
> > The variables in patients' EHR data mean the diagnosis and procedure codes. For example, the diagnosis code reflects the disease the patient has. It also should be a node in your graph. It doesn't mean multiple modalities. I agree with you that you are focusing on the time series topic. But you discuss it in the field of clinical notes, I believe it would be better if you could make full use of the data. Thanks for your response, I will keep my score.

---

### Official Review · Reviewer_yFgU · 2024-07-15

**Soundness:** 3
**Presentation:** 3
**Contribution:** 3
**Rating:** 6
**Confidence:** 3

**Summary:**

Throughout this article, the authors have described an approach (KEDGN) to tackle Irregularly Sampled Medical Time Series. In this task the data is composed of multiple variables represented with time series. The sample rate of a time series can be irregular and two time series can have their samples taken at a different timestamp.
In the approach described the authors propose to first find common knowledge correlation between the variables in the dataset. To do so, they compute a non-linear cosine similarity between embeddings of textual descriptions, obtained using a Pre-trained Language Model.
They then precise this common knowledge correlation by adding a timestamp aware correlation. Finally using this correlation information and a Graph Convolutional Recurrent Network, their approach can predict the class expected of the time series.
They have then analysed the results of their approach first through a benchmark with multiple state of the art approaches. They then added an ablation study and different visual analysis to give a better overview of the behavior of the approach.

**Strengths:**

* The proposal to use Common Knowledge through PLM to find correlation
* Detailed Experiment with a detailed comparison with state of the art approaches
* Detailed exploration of the behavior of the model through an ablation study and visual analysis of the Embedding of the explanations.

**Weaknesses:**

* Details of the Proposed Model could be better explained by giving the intuition behind the equations. (Especially for 4.3.2 and 4.4)

**Questions:**

* Did the authors try to use definitions given by health care professionals to confirm the definition given by the GPT model?
* It is not clear how equation (7) uses equation (6). Indeed, $Z^{(i)}_{v}$ was defined with $i$ being the $i$-th observation point (I believe the $i$-th sample of this specific variable $v$).
But on the other hand the equation (7) uses $Z$ with $t$ being a timestamp. --

It is not clear if $Z^{(i)}_{v}$ is defined for a timestamp in which no observation was made for a variable.
* It is not clear why the equation (9) contains a transpose.

**Limitations:**

It appears the limitations have been discussed in sufficient detail.

---

> ### Author Rebuttal · Authors · 2024-08-06
>
> # Responses to Reviewer yFgU (1/1)
>
> Dear reviewer yFgU,
>
> Thank you for your valuable feedback on the formula details of the article. Here is the response:
>
> **W1. Details of the Proposed Model could be better explained by giving the intuition behind the equations. (Especially for 4.3.2 and 4.4)**
>
> A1. Firstly, we need to correct equation(8) in Section 4.3.2 to:
> $ G^{(t)}_{ij} = A^{(t)}\_{ij} \times (1 - W\_{ij} |D^{(t)}\_{i} - D^{(t)}\_{j}|).$
>
> The intuition behind this equation is as follows: If two variables have similar densities at a given time, their correlation should be higher. Conversely, if the density of one variable increases or decreases significantly at subsequent times, the correlation between these variables will decrease. For example, Figure 6 in Section 5.4.3 shows that NIDiasABP and HR have similar densities at $t=4$, but by $t=15$, the density of NIDiasABP decreases, which reduces its correlation with HR. Therefore, we use the absolute difference in densities between the two variables and add a negative sign so that the density difference is negatively correlated with the correlation.
>
> Regarding the formulas in Section 4.4, they integrate variable-specific parameter spaces and dynamic graph networks into the GCRNN backbone network. Equation (9) represents the standard definition of  $1st$-order Chebyshev polynomial expansion approximation for graph convolution. Equations (10-12) define the operations in GCRNN. Equation (13) indicates that only the hidden states of observed variables are updated at each time step, ensuring that the model fully adheres to the original ISMTS pattern.
>
>
>
> **Q1. Did the authors try to use definitions given by health care professionals to confirm the definition given by the GPT model?**
>
> A2. In this paper, what the model required as input are general medical domain knowledge text descriptions of each variable. These domain knowledge is relatively well-established and fixed, and can be confirmed by consulting medical literature. Therefore, we did not seek help from health care professionals.
>
>
>
> **Q2. $Z^{t}$ 's definition is not clear.**
>
> A3. We apologize for the confusion regarding the notation. We have revised the definition of $Z^{(t)}$ in Equation (6) as follows:
> $$
> Z^{(t)}=Z_{i, v}=
> \begin{cases}
> ((t_{i,v}-t_{i-1,v}) + (t_{i+1,v}- t_{i, v})) / 2, \quad  \text{if }t_{i+1,v} \text{ and } t_{i-1,v} \text{ are both not None.}\\\\
> t_{i,v}-t_{i-1,v}, \quad  \text{if }t_{i+1,v}  \text{ is None.} \\\\
> t_{i+1,v}- t_{i, v}, \quad  \text{if } t_{i-1,v} \text{ is None.}\\\\
> t_{max} / 2, \quad  \text{if }t_{i+1,v} \text{ and } t_{i-1,v} \text{ are both None.}
> \end{cases}
> $$
> Here, $Z^{(t)}=Z_{i, v}$ represents the average density at time stamp $t$ for the $v$-th variable's $i$-th observation point. It is important to note that density is used for dynamically adjusting edge weights, and edges only exist between actual observations. Therefore, when a variable has no observations, there are no edges to adjust, so the density calculation is not needed in such cases. Hence, we did not consider this situation in defining the average density.
>
>
>
> **Q3. It is not clear why the equation (9) contains a transpose.**
>
> A4. As indicated in Equation (4), after computing the cosine similarity between variable node embeddings, we apply the Softmax function to normalize each column to obtain the graph. Since matrix multiplication involves the left matrix’s specific rows and the right matrix’s specific columns, and each column of the right matrix $S \in \mathbb{R}^{V \times I}$ represents the values of all nodes for a specific input channel, we transpose $I_V + G(t) $ in Equation (9). This transposition converts each column into a row, representing the correlations of one variable with all other variables, ensuring that the sum of the output edge weights for each variable node is 1.

---

### Official Review · Reviewer_vHK1 · 2024-07-16

**Soundness:** 3
**Presentation:** 3
**Contribution:** 3
**Rating:** 5
**Confidence:** 5

**Summary:**

This paper investigates the problem of irregularly sampled medical timeseries classification. The core idea is to use PLM to obtain semantic embeddings for variables, which are used to form a variable correlation graph. Then, the variable correlation graph is dynamically adjusted with the observations, based on which a spatiotemporal graph neural network is used to learn timeseries representation for classification. Typical experiments on widely used benchmarks are conducted to evaluate model performance.

**Strengths:**

1. The paper is well-written and easy to understand. The authors clearly tell the story of using variable correlation graph to solve irregularly sampled medical timeseries classification problem.
2. The investigation of different PLMs on variable correlation graph discovery is important and useful to the community, although the conclusion is that most PLMs achieve similar results, and even name can achieve comparable performance with wikipedia.
3. The experiment section is well defined with various results illustrated.

**Weaknesses:**

1. The parameter settings for baselines are not fair. In lines 228-230, the authors simply follow parameters in reference. However, such setting is not a fair way comparison due to several reasons: Some datasets like MIMIC-III for experiments are not public, i.e., although we have the same raw data source, the experimented datasets are never exactly same, and may be very different from distribution and statistics. This can be validated from the comparison between the results of Raindrop on P19 dataset from this paper and Raindrop paper. The performance has a significant difference, which implies the parameters should searched completely. Thus, grid search for baseline parameters is necessary.

2. The choice of GCRNN here is not well clarified. There are many spatiotemporal graph neural networks like STGNN /ST-GCN which can be used here. Moreover, the ablation study of graph convolution and temporal modeling for GCRNN should be conducted to evaluate the performance gains.

3. Some claims or statements intend to emphasize the model performance, which however is obviously set by design itself instead of model learning. For example, in Line 337 the authors state ''no correlation is observed with DiasABPas it has not been observed yet''. Actually, this is not by learning but the initial masking operation of adjacency matrix.

4. An important baseline of irregularly sampled medical timeseries modeling is missed here, i.e., StraTS [1].

5. The ablation study doesn't identify very effective module of the proposed work. For example, Text, KEE and DAG's performance gains are marginal, so are these module necessary?
[1] Self-Supervised Transformer for Sparse and Irregularly Sampled Multivariate Clinical Time-Series, TKDD.

Typos:
1. The symbols or bold texts in Equation (10-12) are wrong.

**Questions:**

1. What's the function of activation function $\sigma$ in equation (7). In my opinion, this activation function reflects the dynamics of variables, like time decay or exponential increase. So, how to determine the function and what are the effects of different activation functions.
2. How is the parameter complexity of the model? Since each variable will require an independent $W_i$, the computation complexity might be very large.
3. This paper emphasizes the importance of using text information of variables. However, the performance of different text sources seems close. Especially, only using Name can achieve similar even better performance. From ablation study in Table 3, we can observe even without textual representations, the model can achieve almost similar performance. Thus, this part looks useless, which is opposite to the commonsense.
4. Is Figure 5(a) calculated from f(E) or E? If in the f(E), the authors should also provide the correlation graph from E, so that we can evaluate whether model learning is effective. To my experience, even bert embedding of these variables can achieve similar correlation graph. Also, why graph (a) is not symmetric? What's meaning of different correlations between (FiO2, HR) and (HR, FiO2)?
5. What's the basic time resolution or sampling rate for each dataset?
6. How is the static features combined with timeseries are not described in paper.
7. How are the input length and prediction length determined are not provided. Usually, in mortality prediction, we use the first 48 hours ICU data as input to predict whether the patient will die in the hospitalization. What's setting for this paper?

**Limitations:**

The authors didn't comprehensively discuss the limitations of this paper, which is not accord with the paper checklist.

---

> ### Author Rebuttal · Authors · 2024-08-06
>
> # Responses to Reviewer vHK1(2/3)
>
> **Q1. The function of activation function in equation (7). **
>
> A1. On one hand, as you mentioned, this activation function reflects the dynamics of variables, such as time decay or exponential increase. On the other hand, this activation function serves a normalization purpose because the edge weights in the knowledge-empowered complete graph $\boldsymbol{A}$ are normalized values. If we directly use the absolute value of the density to adjust the edge weights, the values might become excessively large or small, which would severely disrupt the basic graph structure learned from textual knowledge. Typical activation functions with normalization capabilities include Sigmoid and Tanh. Since Tanh performed better in our experiments, we chose Tanh.
>
> The experimental results for different activation functions are shown in rows W/O $\sigma$ and $\sigma$ Sigmoid in Table 2 of the newly submitted PDF.
>
>
>
> **Q2. The parameter complexity of the model.**
>
> A2. We have calculated the number of model parameters for each dataset, as shown in Table 5 of the newly submitted PDF.
>
> As shown, the parameter count of our model is not particularly high. It is on the same order of magnitude as Warpformer and significantly lower than Raindrop by three orders of magnitude. Although we calculate an independent $W$ for each variable, the total $ W \in \mathbb{R}^{V \times I \times O} $ does not equate to the parameter count. $ W $ is derived from the multiplication of two matrices: the variable embedding matrix $ Q \in \mathbb{R}^{V \times q} $ and the weight matrix $ W \in \mathbb{R}^{q \times I \times O} $. The first matrix is computed from textual embeddings, and only the second matrix belongs to the model parameters. The sizes $ q $, $ I $, and $ O $ are hyperparameters, independent of the number of variables, and typically set to be less than 16. Hence, the parameter complexity of our model is not high. We will include this parameter complexity analysis in Appendix F.2 on computational costs.
>
>
>
> **Q3. The importance of using text information of variables.**
>
> A3. First, as analyzed in Section 5.4.1, the relative distribution of variable embeddings obtained from different text sources via PLM is consistent with the relative distribution of the variables' time series patterns. Both of them reflects the relative distribution of the variables' inherent sense. The embedding space of variables obtained from different text sources may vary in absolute distribution, but the relative distribution should be similar. So the performance from different text sources should also be similar, which aligns with our expectations and proves incorporating text information has a universally valid.
>
> Moreover, as shown in table 4 of the newly submitted PDF, the overall results of the ablation experiments indicate that the introduction of text contributes to performance improvement and is not useless.  More importantly, as demonstrated in the visualizations in Section 5.4.1, if the variable text embeddings are replaced with learnable embeddings, the resulting learned variable embedding space tends to be nearly uniformly distributed, which lacks interpretability. In contrast, using text embeddings ensures that the variable embedding distribution is consistent with domain knowledge, thereby providing the model with strong interpretability.
>
>
>
> **Q4. Calculation and symmetry issues of correlation graph**
>
> A4. Your observation is correct. Figure 5(a) is computed from $g(E)$. It is true that using the BERT embeddings of these variables can achieve a broadly similar correlation figure, but there are subtle differences induced by $ g(\cdot) $. The introduction of $ g(\cdot) $ is necessary as it not only performs feature reduction but also avoids using a completely fixed prior graph. This preserves the model’s ability to adaptively optimize the graph structure based on different data distributions and downstream tasks.
>
> To validate the effectiveness of $ g(\cdot)$, we provide the ablation experiment results on row W/O $ g(\cdot)$ in table 2 of the newly submitted PDF.
>
> Regarding the asymmetry of Figure 5(a): As indicated in Eq.(4), after computing the cosine similarity between variable node embeddings, we apply the Softmax function to normalize each column to obtain the graph. This normalization ensures that the diagonal elements correspond to the maximum values in their respective columns, but not necessarily the maximum values in each row.
>
> We intentionally set the graph to be asymmetric because some variable correlations are directional. For example, a decrease in insulin level may lead to an increase in blood glucose, but an increase in blood glucose does not necessarily lead to a decrease in insulin level. This is why the edge weight from insulin to blood glucose is not equal to the weight from blood glucose to insulin. The same reasoning applies to HR and FiO2.
>
>
>
> **Q5. The basic time resolution or sampling rate for each dataset.**
>
> A5. In ISMTS, there are significant differences in sampling rates among different samples, variables, and periods. As a result, calculating a basic sampling rate for an entire dataset is challenging and sometimes not very informative. Instead, more relevant statistics for ISMTS datasets include metrics such as missing rate and maximum length.
>
>
>
> **Q6. How are the static features combined with time series.**
>
> A6. We apologize for the omission. We follow the approach used in Raindrop to incorporate the static features. Specifically, static features are first mapped to static vectors through a linear layer and then concatenated with the feature vectors of the time series before being input into the classifier.

---

> ### Author Response · Authors · 2024-08-06
>
> # Responses to Reviewer vHK1(3/3)
>
> **Q7. How are the input length and prediction length determined. **
>
> A7. We apologize for the omission. As you mentioned, for the MIMIC-III, P12, and PhysioNet datasets, we use ICU data from the preceding 48 hours as input to predict the mortality during the hospitalization. For the P19 dataset, we use up to 60 hours of ICU data to predict whether sepsis will occur within the subsequent 6 hours. We will include these details in the "Appendix B: Datasets" section of the revised version.
>
>
>
> **Q8. Limitations**
>
> A8. Although our proposed method effectively guides ISMTS modeling through the domain knowledge from text modality, it has some limitations. The backbone of our model is based on an RNN architecture, which inherently has a sequential computation characteristic that can be a bottleneck in terms of runtime. Additionally, our method is specifically tailored for medical applications, and its performance may be limited in other ISMTS applications, such as human activity recognition, where variables may lack domain knowledge and thus cannot generate high-quality text descriptions. We will include a detailed discussion of these limitations in the revised version of the paper.

---

> ### Author Response · Authors · 2024-08-12
>
> Dear Reviewer vHK1,
>
> Hope this message finds you well.
>
> We appreciate the diligent efforts of you in evaluating our paper. We have responded in detail to your questions. As the discussion period will end soon, we would like to kindly ask whether there are any additional concerns or questions we might be able to address.
>
> Once more, we appreciate the time and effort you've dedicated to our paper.
>
> Best regards,
>
> Authors

---

### Author Rebuttal · Authors · 2024-08-06

# Global Response

We sincerely thank all the reviewers for their consistent positive feedback regarding the significance of our work, the novelty of our approach, the thoroughness of our experiments and analyses, and the quality of our presentation. Additionally, we greatly value the reviewers' insightful and constructive comments, which have significantly contributed to enhancing and refining this work. Here, we will response each of the weaknesses and questions raised by the reviewers in order to eliminate reviewers' concerns to the greatest extent possible.

Due to the limited number of pages and characters allowed for rebuttal, we will begin our responses to Reviewer **vHK1** on this global rebuttal page. The responses to reviewers **yFgU, 68iN, and n8KK** can be found below the respective official review pages.



# Responses to Reviewer vHK1(1/3)

Dear reviewer vHK1,

We thank for your carefully reading on our paper and valuable comments on the details of our work, the following is our responses, please download the 1-page PDF document we have newly uploaded at the left lower corner of this rebuttal page for easier reference and reading.



**W1.Grid search for baseline parameters is necessary.**

A1. We performed a grid search across four datasets around the optimal parameter range for the three most recent SOTA baselines due to time constraints. The results are shown in Table 1 of the newly submitted PDF.

As you mentioned, some baselines did indeed show some improvement in performance on certain datasets after the grid search. However, our method still consistently maintained leading performance across all datasets. In comparison, although some baselines demonstrated competitive performance, they still exhibited significant disadvantages on a few datasets. For instance, Warpformer on the Physionet dataset, $DGM^{2}-O$ on the MIMIC-III dataset, and Raindrop on the Physionet dataset. Therefore, the effectiveness of our method remains evident.



**W2. The choice of GCRNN here is not well clarified./The ablation study of graph convolution and temporal modeling.**

A2. Firstly, our main contribution is not the improvement of the GNN backbone network but the learning of variable-specific parameter spaces and dynamic graphs. Therefore, what we need is a simple, effective, and easy-to-adapt backbone GNN for ISMTS. Existing spatio-temporal graph neural networks can be roughly classified based on their temporal module into three main categories: Convolution-Based, Attention-Based, and Recurrence-Based.

- Convolution-Based methods capture temporal dependencies using TCNs. However, the fixed-step kernel size cannot perceive different time spans effectively, making them unsuitable for ISMTS.
- Attention-Based methods face challenges due to the misalignment of sampling times across different variables in ISMTS. Computing variable-specific attention is difficult to parallelize in the variable dimension, and their computational complexity scales quadratically with the sequence length, limiting scalability. Furthermore, attention outputs have the same dimensions as inputs, requiring additional design for extracting final sample features from observation-level representations.
- Recurrence-Based methods update variable latent states based on whether they are observed, which allows parallel computation in the variable dimension and suits ISMTS. The final sample feature representation can be directly obtained from the latent states at the last observation time, making this approach simple, effective, and well-suited for ISMTS.

Thus, we chose GCRNN as the backbone GNN for this paper.

Regarding the ablation study of graph convolution and temporal modeling for GCRNN, we set the graph into an identity matrix and remove the time embedding in the structured encoding respectively. The results are shown in rows W/O GCN and W/O TE in Table 2 of the newly submitted PDF.

The results show that both the introduction of graph convolution and time embedding are necessary.





**W3. Some claims or statements intend to emphasize the model performance, such as Line 337.**

A3. We apologize for the misunderstanding caused by our statement. We have revised the phrasing in Line 337 to: "Around t = 4, HR shows a strong correlation with NIDiasABP, while the correlation with DiasABP is masked as 0 since DiasABP has not been observed yet."



**W4. An important baseline of ISMTS modeling (StraTS) is missed here.**

A4. Thank you for your reminder. We will include StraTS in the "Irregularly Sampled Multivariate Time Series Modeling" section of the related work in future versions. Additionally, we have added comparative experiments with StraTS. The experimental results are shown in Table 3 of the newly submitted PDF.



**W5: The ablation study doesn't identify very effective module of the proposed work.**

A5. We have calculated the average performance improvement brought by the three modules across the four datasets, as shown in Table 4 of the newly submitted PDF.

Overall, all three modules contribute to performance improvement. Given that the baseline values of AUROC are relatively high (typically above 85%), the absolute value of the AUROC improvement may appear lower. In this context, a single module that can achieve an AUROC improvement greater than 0.5% is significant. Additionally, all three modules bring more than 1.5% ↑ in AUPRC, which has been shown to be more sensitive to imbalanced samples. Therefore, we consider these modules essential and indispensable components of our model.



**W6. The symbols or bold texts in Equation (10-12) are wrong.**

A6.  We have removed the bold text styling from the symbols in equations (10-12).
$$
r^{(t)}=\sigma(\Theta_{r} \star_{G^{(t)}} [X^{(t)}|| H^{(t-1)}] + b_{r}),
$$

$$
u^{(t)}=\sigma(\Theta_{u} \star_{G^{(t)}} [X^{(t)}|| H^{(t-1)}] + b_{u}),
$$

$$
C^{(t)}=tanh(\Theta_{C} \star_{G^{(t)}} [X^{(t)}|| (r^{(t)} \odot H^{(t-1)})] + b_{C}),
$$

---

### Decision · Program_Chairs · 2024-09-25

**Decision:**

Accept (poster)

**Comment:**

The paper presents a novel approach to tackle a critical problem in healthcare data analysis: the irregular sampling of time series data, which is common in medical applications. The Knowledge-Empowered Dynamic Graph Network (KEDGN) leverages pre-trained language models (PLMs) to generate semantic embeddings for each variable based on their medical properties. This enables the dynamic modeling of inter-variable correlations that evolve over time. By incorporating variable-specific parameter spaces and dynamically adjusting the variable graph, the proposed model enhances the ability to capture fine-grained temporal dependencies and correlations in ISMTS, significantly outperforming existing methods. This innovative integration of textual knowledge with dynamic graph neural networks represents a meaningful contribution to the field, offering potential improvements in clinical decision-making and predictive modeling.

Furthermore, the proposed work is well-grounded in extensive experimentation on multiple healthcare datasets, showcasing the effectiveness of KEDGN against state-of-the-art methods. The detailed analyses, including an ablation study and visual exploration of the learned variable correlations, provide a comprehensive understanding of the model's capabilities and behavior. It effectively communicates the importance of modeling variable-specific and time-varying correlations in healthcare time series, a topic that is highly relevant to the machine learning and medical communities. The incorporation of medical knowledge through PLMs, combined with the novel graph-based approach, provides a robust framework for future research in irregularly sampled time series modeling.